# A repressor-decay timer for robust temporal patterning in embryonic *Drosophila* neuroblast lineages

Inna Averbukh[1†], Sen-Lin Lai[2†], Chris Q Doe[2*], Naama Barkai[1*]

[1]Department of Molecular Genetics, Weizmann institute of science, Rehovot, Israel; [2]Institute of Neuroscience, Institute of Molecular Biology, Howard Hughes Medical Institute, University of Oregon, Eugene, United States

**Abstract** Biological timers synchronize patterning processes during embryonic development. In the *Drosophila* embryo, neural progenitors (neuroblasts; NBs) produce a sequence of unique neurons whose identities depend on the sequential expression of temporal transcription factors (TTFs). The stereotypy and precision of NB lineages indicate reproducible TTF timer progression. We combine theory and experiments to define the timer mechanism. The TTF timer is commonly described as a relay of activators, but its regulatory circuit is also consistent with a repressor-decay timer, where TTF expression begins when its repressor decays. Theory shows that repressor-decay timers are more robust to parameter variations than activator-relay timers. This motivated us to experimentally compare the relative importance of the relay and decay interactions in vivo. Comparing WT and mutant NBs at high temporal resolution, we show that the TTF sequence progresses primarily by repressor-decay. We suggest that need for robust performance shapes the evolutionary-selected designs of biological circuits.
DOI: https://doi.org/10.7554/eLife.38631.001

**\*For correspondence:**
cdoe@uoneuro.uoregon.edu (CQD);
naama.barkai@weizmann.ac.il (NB)

[†]These authors contributed equally to this work

## Introduction

Multicellular organisms shape their body plans during embryonic development through parallel processes that occur at different spatial positions and at different times. Spatial coordination of patterning depends on direct cell-cell communication and on long-range signaling by secreted morphogens. By contrast, temporal coordination is often achieved through cell-autonomous processes that measure time-delays (*Pourquié, 1998*). Molecular circuits implementing biological timers have been described (*Murray, 2004*; *Reppert and Weaver, 2002*; *Simon et al., 2001*), but the basis for their robust functioning is not well understood.

Biological timers play a key role in central nervous system (CNS) development of invertebrates and mammals, where a small pool of progenitors generates a vast amount of neuronal diversity (*Grosskortenhaus et al., 2005*; *Isshiki et al., 2001*; *Kambadur et al., 1998*; *Kohwi and Doe, 2013*; *Kohwi et al., 2011*; *Li et al., 2013*). In *Drosophila*, NBs delaminate from a ventral neuroectoderm at embryonic stages 9–11 to form an orthogonal two-dimensional grid with 30 NBs per hemi-segment. After formation, each NB undergoes asymmetric cell divisions every ~45 min to produce a series of ganglion mother cells (GMCs), each of which divides into two post-mitotic neurons. The identity of each neuron is determined by the spatial position of the parental NB and by the TTF it inherits from the NB at the time of birth (*Doe, 2017*; *Kohwi and Doe, 2013*; *Pearson and Doe, 2004*). Temporal information is therefore conveyed by the NB cell-intrinsic timer, which drives the sequential expression of four TTFs: Hunchback (Hb), Krüppel (Kr), Pdm (Flybase: Nubbin and Pdm2), and Castor (Cas) (*Brody and Odenwald, 2000*; *Isshiki et al., 2001*; *Kambadur et al., 1998*).

The molecular basis of the NB TTF timer has been characterized: Hb expression is initiated by an external signal, but subsequent dynamics depends on cross-regulation between the TTFs themselves (*Cleary and Doe, 2006*; *Grosskortenhaus et al., 2006*; *Isshiki et al., 2001*; *Tran et al., 2010*). In addition, the orphan nuclear hormone receptor, Seven-up (Svp), is required to switch off Hb expression, but not for specifying early neuronal fates (*Kanai et al., 2005*; *Kohwi et al., 2011*; *Mettler et al., 2006*; *Tran et al., 2010*).

The TTF expression sequence is largely independent of the cell cycle: some NBs undergo just one cell division during the Hb expression window, whereas others undergo two or three divisions (*Baumgardt et al., 2009*; *Doe, 2017*; *Isshiki et al., 2001*) indicating that cell division does not direct this temporal transition within the TTF cascade. Consistent with that, mutations that arrest the cell cycle still allow normal TTF expression from Kr onward. Also, NBs can undergo the entire expression sequence when cultured in isolation in vitro indicating TTF timer progression is independent of overall embryo development (*Brody and Odenwald, 2002*; *Grosskortenhaus et al., 2005*; *Kambadur et al., 1998*). While progression along the transcription cascade is largely independent of the cell cycle, these two processes must remain synchronized in order to ensure reproducible NB lineage. How do embryos limit temporal variations that could desynchronize these two largely independent processes?

We hypothesize that reliable progression of the TTF cascade derives from the cell-intrinsic ability of the TTF timer to buffer variation (noise) in its molecular parameters. We therefore used mathematical modeling to analyze theoretically the robustness of the timer mechanism. The TTF-timer is commonly described as a relay of activators (*Doe, 2017*; *Rossi et al., 2017*). Its regulatory circuit, however, contains also regulations compatible with a repressor-decay timer, in which TTF starts expressing when its repressor decays to a sufficiently low level. The relative contributions of the activator-relay and the repressor-decay interactions to TTF progression depend on unknown molecular parameters

Our computational analysis revealed that the two mechanisms show different sensitivity to variations in their molecular parameters, with the decay-timer being significantly more robust than the relay timer. This finding motivated us to examine experimentally the relative contribution of the relay and decay interactions to the in-vivo progression of TTF expression. This was done by measuring the TTF dynamics in wild-type and mutant embryos at high temporal resolution. Our results show that removing a relay-controlling factor has limited consequences on the time of induction of the controlled TTF compared to the removal of a decay-controlling factor, indicating that the decay-based interactions dominate the progression of the in vivo TTF timer. We conclude that NB temporal patterning in *Drosophila* is driven by a timer whose progression is dominated by repressor decay, while activator relay plays a minor role. We propose that the increased robustness of this decay timer, as suggested by our computational analysis, is critical for ensuring the synchrony of this timer with the parallel, yet independent, progression of the cell division cycles.

## Results

### The TTF regulatory circuit combines activator-relay and repressor-decay interactions

The sequential expression of TTFs within the dividing NB is commonly described as a relay of activators, in which each activator accumulates until reaching a threshold needed for inducing the next activator in the cascade (*Doe, 2017*; *Rossi et al., 2017*). Examining the TTF regulatory circuit, however, we noted that in addition to these activator-relay interactions, the regulatory circuit includes two additional interaction types: backward interactions, whereby a TTF inhibits the expression of a TTF upstream in the cascade, and repressor-decay interactions, whereby an upstream regulator represses the expression of a downstream TTF (*Figure 1A–B*). Therefore, at least in principle, the TTF timer can progress not only through a relay of activators, but also through decay of repressors, where target genes are induced once a repressor decays below some threshold level.

The activator-relay and repressor-decay regulations could both contribute to the progression of the TTF timer. Alternatively, one timer type could dominate. The relative contribution of the different regulations to the initiation of TTFs expression depends on the in vivo parameters, whose values

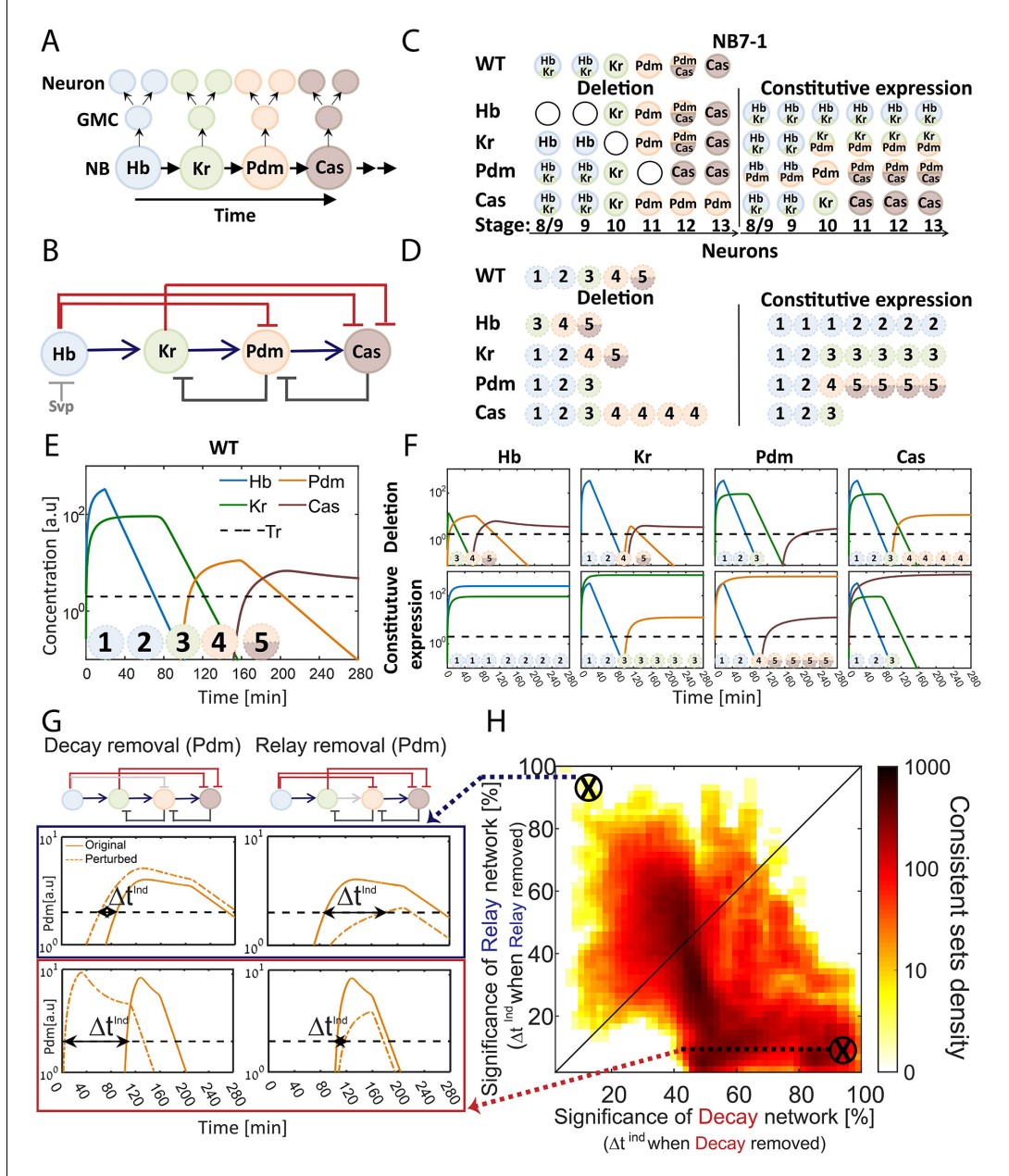

**Figure 1.** The TTF regulatory circuit combines activator-relay and repressor-decay timers. (A) The embryonic neuroblast TTF timer: In the *Drosophila* embryo, NBs express four TFs in a temporal sequence, as shown. We term this sequence the TTF timer. NBs divide asymmetrically to generate GMCs, whose further divisions produce post-mitotic neurons. The identity of both the GMCs and the post-mitotic neurons depends on the TTF expressed at the time of NB division, as shown in panel C, top left. (B) The TTF regulatory circuit: experimentally defined cross-regulation between TTFs include interactions that propagate the cascade through activator relay (blue) or repressors decay (red), and backward interactions (black). (C) Summary of TTF expression in NB7-1 of wild type, mutant, and misexpression genotypes. TTFs are color-coded as in (A). Developmental stages indicated at the bottom and genotypes on the left. Data from (***Grosskortenhaus et al., 2006***; ***Isshiki et al., 2001***; ***Tran et al., 2010***). (D) Summary of neuronal identity in the NB7-1 lineage of wild type, mutant, and misexpression genotypes. Data from (***Grosskortenhaus et al., 2006***; ***Isshiki et al., 2001***). (E) TTF timer model reproduces all reported phenotypes: a model was formulated that includes all regulatory interactions shown in ***Figure 1B***. Shown are the simulated dynamics of TTFs expression for a set of experimentally consistent parameters. Progeny identity was defined by the TTFs whose expression exceeded a constant threshold (dashed line) at the time of division (see ***Supplementary file 2*** for mapping of expressed TTFs to neuronal fates). (F) Mutant (top) and constitutive expression (bottom) models using the same parameter set as in 1E (see Materials and methods for details). (G–H) Consistent circuits are distributed through the Decay-Relay timer space: over $10^6$ circuits differing by parameters choice were considered. A subset of ~$10^5$ circuits reproduced all experimentally defined phenotypes and are referred to as consistent circuits, or consistent parameter sets. Each consistent circuit was positioned on the Decay-Relay plot (H) based on the changes in Pdm and Cas induction (ind) times following removal of the respective activator-relay

*Figure 1 continued on next page*

*Figure 1 continued*

or repressor-decay interactions (grey bars, top of G). The density of consistent circuits in the Decay-Relay plot is color-coded. Note the larger density of consistent circuits in the regime of decay-timers. See Materials and methods for details.

DOI: https://doi.org/10.7554/eLife.38631.002

are not known and are difficult to measure. We therefore examined whether the in vivo parameters can be distinguished computationally.

We formulated a model of the TTF timer that includes all experimentally described interactions, capturing their strengths by 17 independent parameters. Main parameters in this model include TTF degradation rates, TTF production rates, and the expression thresholds defining the minimal TTF expression levels required for activating or repressing the controlled TTF and to confer neuronal identities (*Supplementary file 1*). The model is summarized by four ordinary differential equations (ODEs), whose solution simulates the temporal dynamics of the four TTFs. Note that model dynamics depends on the choice of the kinetics parameters (*Supplementary file 1*). Further, varying parameters qualitatively changes the regulatory network by varying the relative influence of the different interactions on TTF temporal dynamics. This allows us to capture a wide range of networks, ranging from decay dominant ones, through any mixed models, to relay dominant networks all the while taking into account all experimentally observed TTFs and cross-regulations (*Supplementary file 1*).

To define parameters consistent with the in vivo dynamics, we compiled phenotypes of mutant embryos, focusing on the best characterized NB7-1 lineage (*Figure 1C–D*). Available data describes which TTF(s) are expressed by the dividing NB, within the GMCs, and by the post-mitotic neurons (*Grosskortenhaus et al., 2006*; *Isshiki et al., 2001*). This data is of low temporal resolution and cannot be used to define the precise durations at which each TTF is expressed. Still, this data provides the basic constraints with which our model should comply: the temporal sequence at which TTFs are expressed in our simulations should be consistent with the temporal sequence for wild type, mutant, and misexpression embryos as emerges from reported data (*Figure 1C–D*, *Supplementary file 2*). We therefore define consistent parameter sets as parameter sets for which the solution of model equations meets the above requirements by exhibiting the correct TTF expression sequence in WT and all mutants.

Screening a wide range of parameters, each defining a different circuit, we detected a large number of parameter sets that were consistent with the reported phenotypes (Materials and methods, *Figure 1E–F*). To examine if these consistent sets favor an activator-relay or a repressor-decay timer, we focused on the last two TTFs (Pdm and Cas) for which activating and repressing interactions were described (*Figure 1B*). We examined how their induction time changes when specifically removing either the activator-relay or the repressor-decay interaction (*Figure 1G*). These values – the changes in TTFs induction times when removing either the activator-relay or the repressor-decay interactions – were used to calculate a decay significance and relay significance score for each parameter set (Materials and methods). Based on these significance scores for the two respective timers, we positioned each consistent circuit within a Decay-Relay timer space (*Figure 1H*). In this analysis, we removed the repression of Cas by its immediate repressor (Kr), but left the repression by Hb intact, reasoning that Hb had already decayed at the time of Cas expression.

We observed a higher density of consistent circuits in the region of repressor-decay timers. However, a large number of consistent circuits were equally dependent on both the relay and the decay interactions, and some were driven primarily by relay (*Figure 1H*). We conclude that the available data is not of sufficient resolution to distinguish whether the in vivo parameters progress the TTF timer through an activator-relay or a repressor-decay timer.

## A repressor-decay timer is more robust than an activator-relay timer

To determine whether an activator-relay or repressor-decay timer best explains the observed, we measured the robustness of consistent circuits, namely their sensitivity to variations (noise) in their biochemical parameters. We expected this sensitivity to differ between different circuits, and perhaps also between the timers whose progression is dominated by the activator relay, or repressor-decay interactions. We previously compared the robustness of a one-step timer encoded by an activator accumulation or a decay of a repressor (*Rappaport et al., 2005*). This comparison was done

by considering circuits that (1) have the same decay rate and (2) require the same time transitioning between two given thresholds, thereby controlling for the effects of changing kinetic constants. As we showed, the repressor-decay timer necessarily shows higher robustness under these conditions (*Rappaport et al., 2005*). The reason for this differential robustness is easily appreciated: the time at which a protein decays between two thresholds is only moderately (logarithmically) sensitive to the values of these thresholds (*Figure 2A–C*). In contrast, the time to increase protein levels between two thresholds depends at least linearly, and typically significantly stronger, on thresholds values (*Figure 2A–C*)

We measured the robustness of each of our consistent circuits identified above by scoring the ability of the circuit to buffer the temporal durations of TTF expression phases against moderate (~20–40%) variations in TTF production rates (*Figure 2D and E*, *Figure 2—figure supplement 1*) (see Materials and methods). In our model, introducing fluctuations in TTF production rate captures also fluctuations in thresholds values, as circuit function depends on the ratio of TTF levels to their activation thresholds. Finally, degradation rates were kept constant, as they define the time scale of timer dynamics and equally affect all circuits (*Rappaport et al., 2005* and data not shown).

To rigorously distinguish whether robustness correlates with a specific timer type, we considered again the positioning of all circuits in the Decay-Relay timer space (c.f. *Figure 1H*). Unlike the simple case of the one-step timer, here we did allow some range of possible dynamics, to account for the low temporal precision of the experimental data. This does not affect our ability to compare the consistent circuits since these differences in dynamics are not correlated with location in Decay-Relay space (*Figure 2—figure supplement 2*). High robustness scores were found in the region of repressor-decay timers, while activator-relay timers were significantly less robust (*Figure 2E*, *Figure 2—figure supplement 1*). We conclude that also in the context of the full TTF cascade model, robustness is improved when progressing through repressor decay rather than activator relay.

## A TTF circuit can be positioned in the Decay-Relay timer space based on TTF-deletion phenotypes

The difference in the robustness of the activator-relay and the repressor-decay timer designs motivated us to examine which of the respective interactions dominates in defining the in vivo timer progression. We therefore searched for experiments that could distinguish properties of the in vivo timer.

Experimentally, TTF deletion is an accessible perturbation. As described above, the consequences of such perturbations were previously reported, but at a resolution that was too low to distinguish between the two timer types. Our simulations pointed to one limitation of existing data: for some mutants, the consequence of TTF deletion was defined by measuring the fates of the post-mitotic neurons, and therefore did not provide conclusive data about possible co-expression phases in which two consecutive TTFs are expressed within the NB, but one of them dominates in generating neuronal identity. This significantly limited our ability to precisely deduce the TTF expression timing in either wild-type or mutant embryos, and thereby greatly increased the spectrum of circuits that were scored as consistent with measured phenotypes.

With this in mind, we examined computationally whether the consequences of TTFs deletions, if analyzed at higher resolution, could distinguish between the repressor-decay and activator-relay timers. First, we examined how Pdm induction time changes following deletion of either Hb (its repressor, allowing Pdm induction only after its levels are sufficiently reduced) or Kr (its activator, allowing Pdm induction only after its levels are sufficiently induced). Specifically, for each of the consistent circuits described in *Figures 1H* and *2E* above, we tested how Pdm induction time changes when simulating the removal of Hb and when simulating the removal of Kr (*Figure 3A*). These two values uniquely position each consistent circuit (*Figure 3B*). Once positioned, we color each circuit by its robustness score for the WT circuit (same color as in *Figure 2E*), which allows us to observe robustness of WT circuits as function of their Pdm induction sensitivity to Hb and Kr deletions (*Figure 3B*). As can be appreciated, high robustness was found exclusively in circuits for which Pdm induction was dependent only on Hb decay. Analogous analysis of Cas induction time shows a similar, although less pronounced bias (*Figure 3—figure supplement 1*).

Deletion of the TTF which functions as an activator or repressor abolishes the respective relay or decay interactions. However, it may have additional effects that are not directly related to these interactions. We therefore used our simulations to examine whether TTF deletion phenotypes can

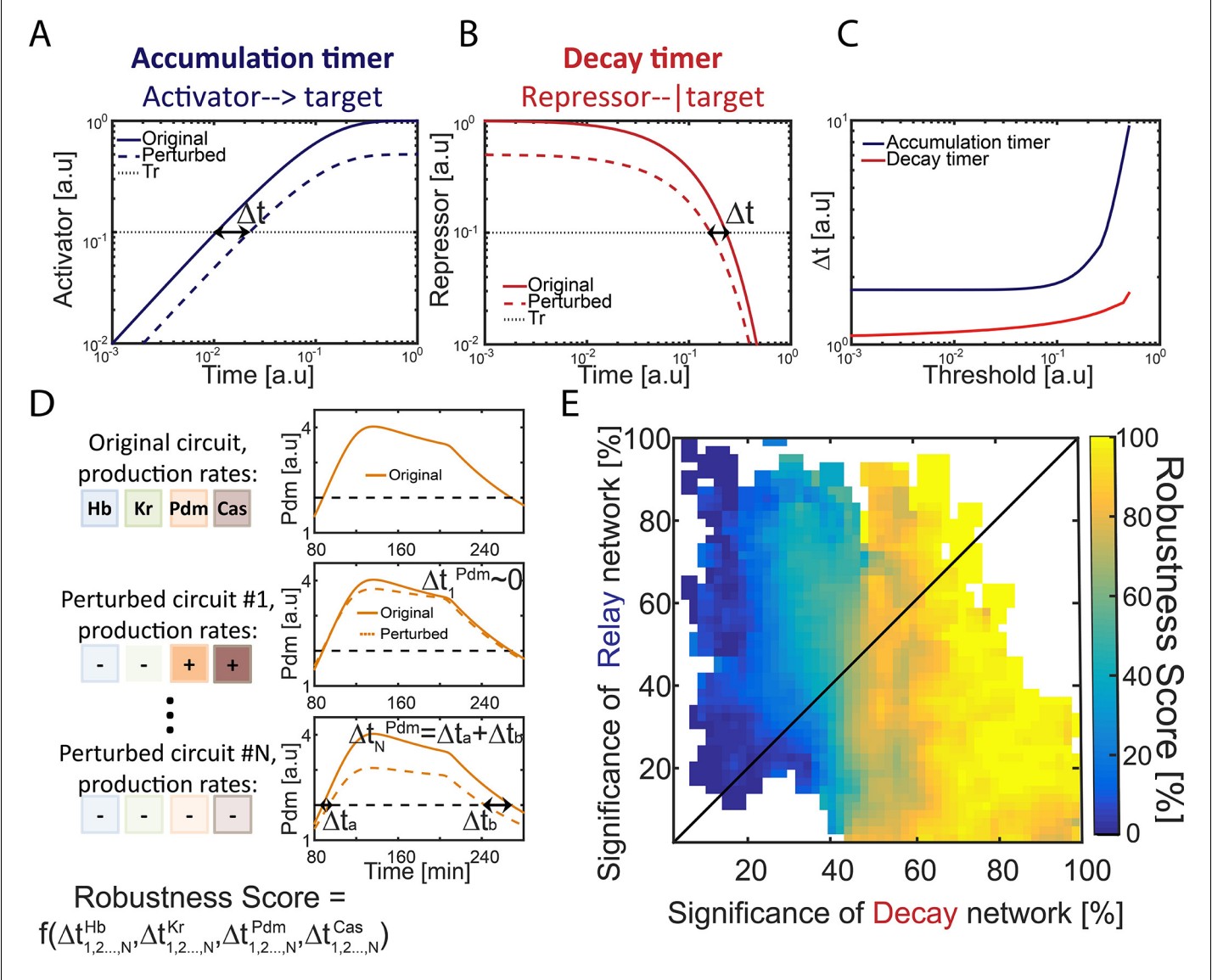

**Figure 2.** Repressor-decay timer is more robust than an activator-relay timer. (**A–C**) Robustness of single-step timers: a single-step timer can be implemented by the accumulation of an activator (**A**) or by the decay of a repressor (**B**). In the accumulation of an activator scenario (**A**), activator production is initiated at t = 0. Once it accedes the threshold Tr, target genes are induced. In the decay of a repressor scenario (**B**), production of a repressor is stopped at t = 0. Once repressor levels have decayed below Tr, target genes would no longer be inhibited. The temporal dynamics of the regulatory proteins are shown in (**A–B**) for reference parameters (solid line) and following two-fold reduction in regulator production rate (dashed line). Timer output is defined by the time-delay from the onset of the dynamics until the regulator reaches the indicated threshold Tr. The change in this output following two-fold change in production rate is indicated (black double arrow), and is shown in (**C**) for different threshold values. See also analytical analysis in (*Rappaport et al., 2005*). (**D**) Illustration of robustness score calculation. Parameter sets were scored for robustness by measuring timer sensitivity to moderate (20%) variations in production parameters (see Materials and methods). When testing set robustness, we solved model equations for all combinatorial combinations of adding or subtracting 20% to all the TTFs production rates. For each noise combination, we compared all TTF expression phase durations to those of the original set solution. If a noise combination yielded phase durations which are all within 10% distance of the respective original durations, the noise combination was considered 'close' to the original. A robustness score was then calculated as the percentage of 'close' noise combinations. An example for a specific parameter set: left- production rates color coded by TTF, right – simulation of Pdm dynamics. Upper: original parameter set, below: two noise combinations:#1 and #N (- sign and lighter shade for reduced production rate, + sign and darker shade for increased production rate). For #1, the perturbed dynamics are 'close' to the original while for #N, a substantial error in phase duration, $\Delta t_N^{Pdm}$, occurred and dynamics are not 'close'. Formula at the bottom of the panel indicates that the robustness score for the original set is a function of all noise combinations 1,2,...,N and durations of all TTF expression phases. (**E**) Distribution of robust circuits in the Decay-Relay timer space. Consistent circuits were positioned on the Decay-Relay plot based on both Pdm and Cas induction, as is *Figure 1H*, and their robustness scores (see Materials and methods), averaged over closely positioned circuits, were color-coded. Considering the positioning of all circuits in the Decay-Relay timer

*Figure 2 continued on next page*

*Figure 2 continued*

space based on Pdm and Cas separately (*Figure 2—figure supplement 1*) and based on both TTFs combined (*Figure 2E*), high robustness scores were found in the region of repressor-decay timers.

DOI: https://doi.org/10.7554/eLife.38631.003

The following figure supplements are available for figure 2:

**Figure supplement 1.** Robustness in decay-relay space separated by TTF.

DOI: https://doi.org/10.7554/eLife.38631.004

**Figure supplement 2.** Expression durations for Pdm (A) and Cas (B), divided by total simulation time (same for all parameter sets).

DOI: https://doi.org/10.7554/eLife.38631.005

predict the consequence of specifically abolishing the respective activator-relay or repressor-decay regulation. To this end, we considered again all consistent circuits. For each consistent circuit, we examined how Pdm induction time changes when specifically removing either the activator-relay (Kr-to-Pdm) or the inhibitor-decay (Hb-to-Pdm) interactions. This allowed us to compare, for each consistent circuit, the change in Pdm induction time when an upstream TTF (e.g. Kr) was deleted, or when the respective interaction (e.g. Kr-to-Pdm) was specifically removed (*Figure 3C,D*). As can be seen, the consequences of these two perturbations were tightly correlated, in particular when comparing the effect of deleting Kr to the removal of Kr-dependent regulation (*Figure 3D*). In the case of Hb, the correlation was somewhat lower, reflecting the multiple roles of Hb in this circuit. However, for the robust parameter sets, which we predicted to describe the in-vivo circuit, correlation was higher (*Figure 3C*). A tight correlation was also observed when comparing the change in Cas induction time following the deletion of its activator (Pdm) or the removal of the Pdm-to-Cas activating link only, and when comparing the consequences of deleting the Cas repressor Kr to the specific removal of the Kr-to-Cas repression link (*Figure 3E,F*). We conclude that following Pdm and Cas expression timing in mutant embryos has the power to inform us not only about the robustness of the in vivo timer, but also about the relative contributions of the relay and decay regulations to the TTF progression.

## Timing of Pdm and Cas expression is highly sensitive to deletion of TTF repressors, but less sensitive to deletion of TTF activators

As described above, our modeling suggests that activator-relay and repressor-decay timers can be distinguished based on the TTF deletion phenotypes, but this would require data on TTF expression levels and timing at a much higher temporal resolution than previously obtained data. To this end, we stained embryos for the TTF of interest, and for the Worniu and Engrailed markers, which allow us to unambiguously identify NB7-1 (*Figure 4A*). TTF protein intensity levels were quantified using confocal microscopy (*Figure 4B*). Variability in staining intensities was controlled by normalizing TTF staining to that of Engrailed, which is constantly expressed in NB7-1.

Our data confirmed the sequential expression of Hb, Kr, Pdm, and Cas within the NB7-1 lineage (data not shown). It further revealed that Pdm expression was longer than expected: about 180 min (*Figure 4C*), a time window that is long enough to generate more than the two previously reported Pdm +GMCs (*Isshiki et al., 2001*). To determine if there were additional Pdm +GMCs in the lineage, we used the NB7-1-specific Gal4 driver to drive the expression of membrane-tethered superfold GFP and co-stained for Pdm and the GMC marker Asense (*Figure 4D*). We found that Pdm was upregulated when NB7-1 was producing the 4th GMC and downregulated after the NB generated the 7th GMC (*Figure 4E*), indicating that four GMCs are produced within the Pdm expression window. Consistent with this finding, we observed a novel Kr$^+$Pdm$^+$ GMC in the lineage, which was not previously reported (*Isshiki et al., 2001*). The newly discovered GMC may produce an Eve-negative motor neuron, interneuron, or undergo programmed cell death (see Discussion).

The Pdm expression window is therefore significantly longer than the duration inferred from the previous data used to calibrate our model. This did not affect our analysis, however, since in our original analysis we did allow for solutions of similarly long Pdm windows, accounting for the uncertainty of the experimental data (*Figure 4—figure supplement 1*). We next used the high temporal resolution expression data to determine whether the relay-timer or decay-timer could best account for Pdm and Cas expression timing in TTF mutant backgrounds. We found that *Kr* mutants did not

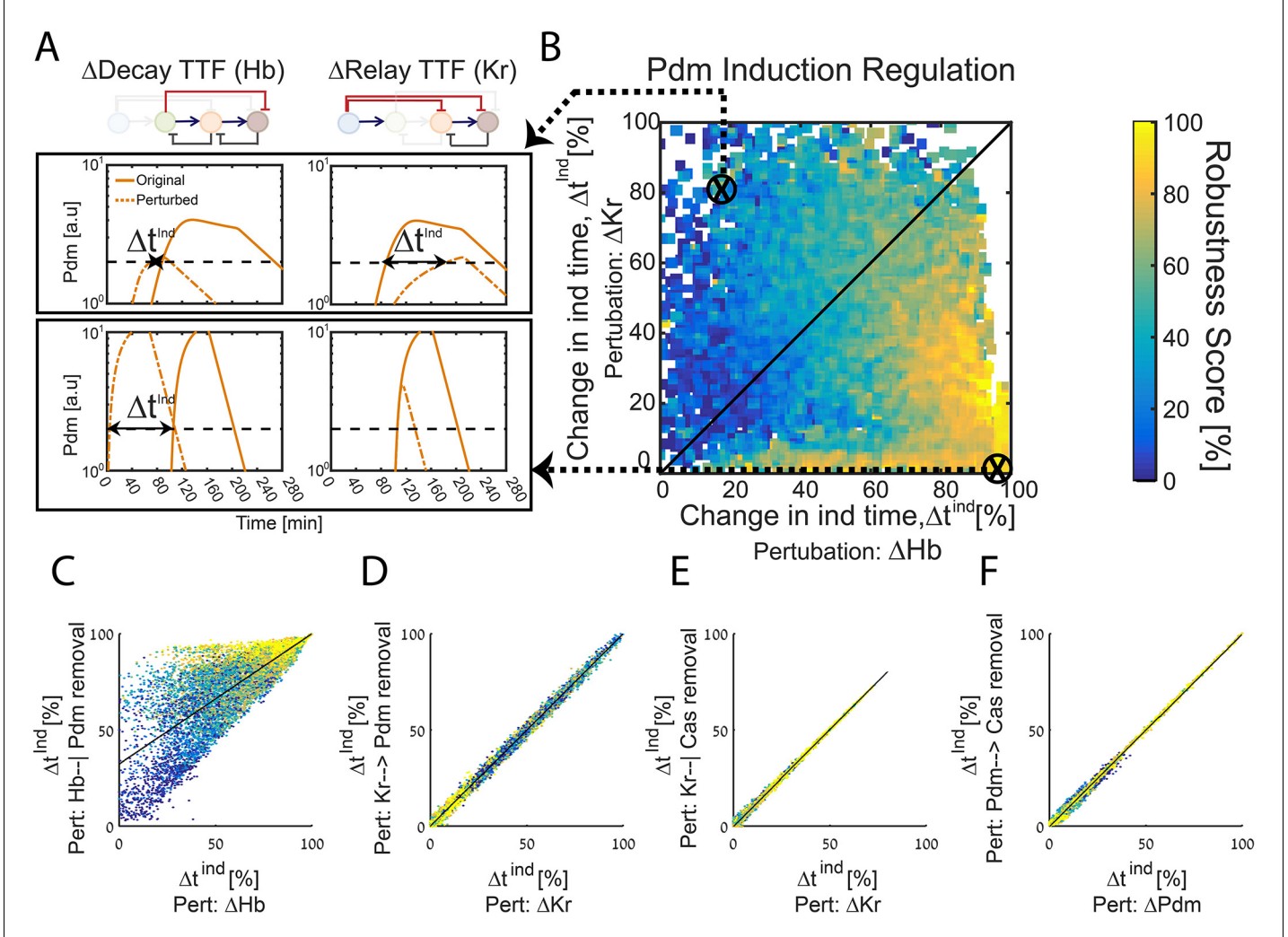

**Figure 3.** The TTF circuit can be positioned in the Decay-Relay timer space based on TTF deletion phenotypes. (A–B) TTF deletion phenotypes can distinguish robust circuits: all consistent circuits, as described in *Figures 1–2* above, were considered. Each consistent circuit was scored by measuring the change in Pdm induction times following deletion of Hb (A, left) or deletion of Kr (A, right). These two values define the 'Hb deletion' and 'Kr deletion' sensitivity of Pdm induction respectively, for each parameter set. We then use these two values to define the Hb-Kr deletion sensitivity space, where we uniquely position each consistent circuit (B). Color-coding circuits based on their robustness score for the WT circuit (as in *Figure 2E*), allows us to observe robustness of WT circuits as function of their Pdm induction sensitivity to Hb and Kr deletions. This analysis shows that robust circuits are only found in a small region in the Kr-Hb sensitivity space, in which Pdm induction time is much more sensitive to Hb than to Kr deletion. (C–F) Sensitivity to TTF deletion (X axis) correlates with the sensitivity to the specific removal of the respective activator-relay or repressor-decay interactions (Y axis): all consistent circuits, as described in *Figures 1–2* above, were considered. For each consistent circuit, the changes in Pdm or Cas induction times following TTF deletion or removal of regulatory interactions was measured. Correlations between the effects of TTF deletion and removal of the respective regulatory link are shown. Each dot in these correlation figures represent one consistent circuit, color-coded by its robustness score for the WT circuit.

DOI: https://doi.org/10.7554/eLife.38631.006

The following figure supplement is available for figure 3:

**Figure supplement 1.** All consistent circuits, as described in *Figures 1–2*, were considered.

DOI: https://doi.org/10.7554/eLife.38631.007

alter Pdm induction timing, whereas *hb* mutants advanced Pdm induction by about two cell-cycles (*Figure 5A–C*), showing that Pdm induction is more sensitive to deletion of the upstream repressor Hb. Similarly, *pdm* mutants did not have as much effect on Cas induction as did *Kr* mutants, showing that Casinduction is more sensitive to deletion of the upstream repressor Kr rather than the upstream activator Pdm (*Figure 5D–F*).

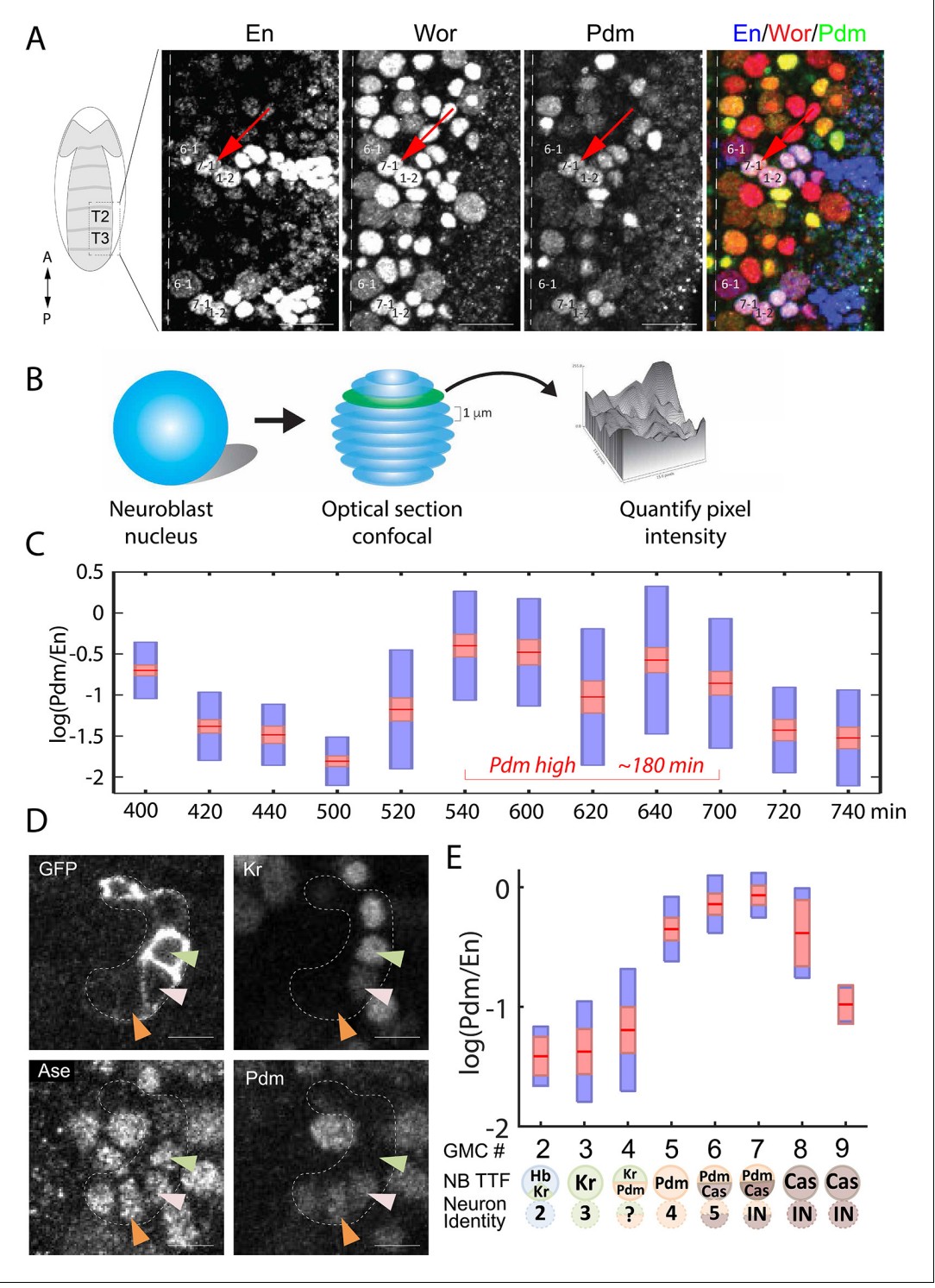

**Figure 4.** High temporal resolution analysis of Pdm expression. (**A**) Confocal image of the neuroblast layer from ventral nerve cord segments T2 and T3 of an early stage 11 embryo (boxed area in illustrated embryo on the left). The NB7-1 was identified by the pan-NB marker Worniu (Wor) and the NB spatial marker Engrailed (En). NB6-1 is the most anterior medial En +NB, with the En +NB7-1 just posterior and lateral (red arrow), and En +NB1-2 completing the diagonal. Genotype: $y^1 w^1$. Scale bar: 20 um. (**B**) Methodology for obtaining transcription factor levels in G1/G2/S-phase of NB7-1. Confocal section stacks (at a 1 um interval) of individual NB nuclei were obtained, and the area and signal intensity of each section were measured and summed to obtain the total intensity of TTFs. (**C**) Data from the number of NB7-1 indicated in *Supplementary file 3* is summarized by box

*Figure 4 continued on next page*

*Figure 4 continued*

plots of measured log(Pdm/En) staining intensity as a function of time. The 1.96 SEM (95% confidence interval) is shown as a red rectangle with a horizontal red line for the mean, with a blue rectangle marking the limits of one standard deviation above and below the mean. Time duration of the Pdm phase (approximately 180 min) is indicated in red. We do not yet have an explanation for the moderate transient decrease in Pdm levels around 620 min. (D) Confocal image of a NB7-1 lineage marked with GFP (*NB7-1-Gal4, UAS-GFP*) in an early stage 11 embryo. Arrowheads indicate three consecutive GMCs (Ase$^+$) which are Kr$^+$Pdm$^-$ (green), Kr$^+$Pdm$^+$ (pink), and Kr$^-$Pdm$^+$ (orange). Genotype: *ac-VP16$^{AD}$ gsb-Gal4$^{DBD}$ UAS-superfoldGFP*. Scale bar: 5 um. (E) Data from the number of NB7-1 indicated in **Supplementary file 3** is summarized by box plots of measured log(Pdm/En) staining intensity as a function of the number of progeny GMC. The 1.96 SEM (95% confidence interval) is shows in red rectangle with a horizontal red line for the mean, one standard deviation above and below the mean in blue rectangle. Scheme below the X axis shows which neuronal progeny is hypothesized to be derived from the GMC.
DOI: https://doi.org/10.7554/eLife.38631.008

The following figure supplement is available for figure 4:

**Figure supplement 1.** In color: the number of Pdm positive neurons predicted to be created for each consistent parameter set, averaged over close parameter sets.
DOI: https://doi.org/10.7554/eLife.38631.009

Cas induction in *Kr* mutants was moderate compared to its induction in WT embryos. This moderate induction is sufficient to induce the Cas-positive U5 neuron, produced in *Kr* mutants (**Isshiki et al., 2001**), and may result from its earlier induction, at a time when Hb, its second repressor, is still highly expressed. Consistent with the contribution of Hb to Cas repression, Cas induction was advanced in Hb mutants to stage 11, similarly to *Kr* mutants (**Figure 5—figure supplement 1**).

Our measurements defined the change in Pdm and Cas induction times following the deletion of their activator or repressor TTFs. Both induction times showed higher sensitivity to repressor deletion than to the deletion of their activator, suggesting that the in-vivo timer progresses primarily through the decay of repressors, with the activator relay playing a minor part.

To examine this conclusion more rigorously, we have linked the measurements to our theory, by examining which of the consistent circuits, identified previously, are also consistent with the higher resolution data we obtained. As can be seen in **Figure 5—figure supplement 2**, our higher resolution data indeed largely restricted the range of possible parameters that remained consistent. This allowed us to more precisely position the in vivo circuit on the Decay-Relay space (**Figure 5G–H**, Materials and methods). As can be seen, the in vivo circuit is in the region of robust timers that are dominated by repressor-decay interactions (**Figure 5I**). We therefore conclude that the timing of TTF expression is driven by a repressor-decay mechanism, rather than an activator-accumulation mechanism (**Figure 5j**).

## Discussion

Our study suggests that a repressor-decay timer drives the sequential TTF expression to generate stereotyped temporal fate specification in *Drosophila* embryonic NB lineages (**Figure 5J**). This finding may appear surprising, as previously this timer was thought to progress through a relay of activators achieved by feed-forward activation combined with feedback repression. A similar activator-relay mechanism was implicated also in driving the TTF cascade in *Drosophila* optic lobe NB lineages (**Bertet et al., 2014**; **Li et al., 2013**). Still, experimentally established cross-regulations between the TTFs are consistent with both activator-relay and repressor-decay mechanisms, and the parameters defining the relative contributions of repressor decay or activator accumulation to the expression timing of each TTF were unknown.

We provided evidence that the TTF progression is dominated by the repressor-decay interactions by measuring the expression timing of the last two TTFs in the cascade, Pdm and Cas. In both cases, TTF induction time was defined by the reduction in upstream repressor level, but showed little, if any change when upstream activator was deleted. Kr induction was not included in this analysis since we found that Kr was maintained in *hb* mutant embryos (data not shown (**Isshiki et al., 2001**)), indicating that Kr is induced by a factor external to the cascade, similarly to Hb. Notably, Kr remained constitutively expressed in embryos which were forced to express constitutive levels of Hb.

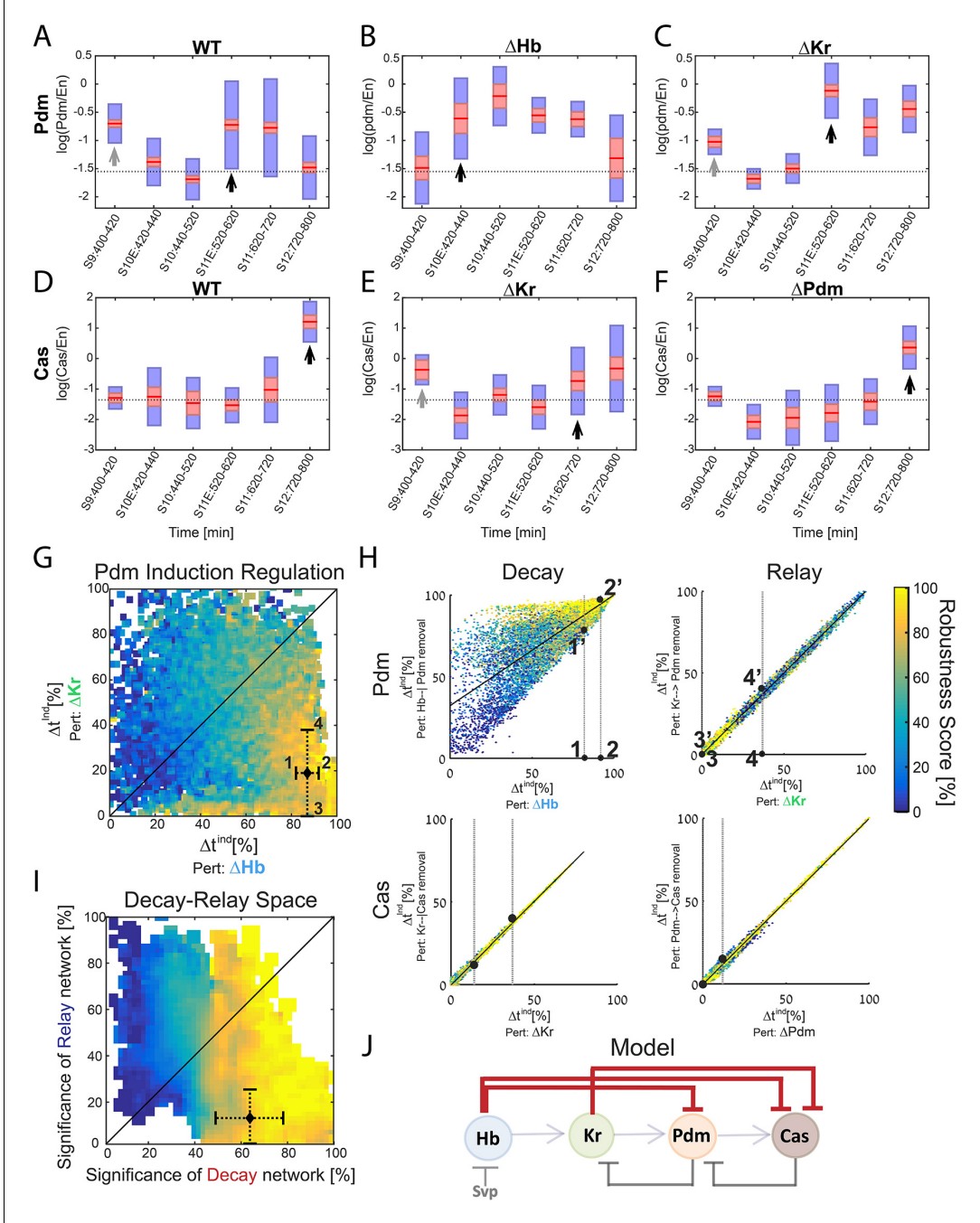

**Figure 5.** Positioning the in-vivo TTF circuit in the Decay-Relay timer space. (A–C) Pdm levels in NB7-1. Pdm levels was quantified in wild-type and mutant embryos. Data from the number of NB7-1 indicated in *Supplementary file 3* is summarized by a box-plot, with 95% confidence interval shown in red rectangle, mean in red horizontal line and the one SD above and below the mean in blue. Developmental stages are indicated on the X axis. Stages of Pdm induction are indicated by black arrows. Induction stages were determined by calculating a background Pdm level as the mean of Pdm in the second and third stages (S10E and S10), in WT. The stage of induction was defined as the first stage for which the 1.96 SEM (95% confidence interval- red rectangle) was above background (see Materials and methods for details). Early transient inductions are indicated by grey arrows. (D–F) Cas levels in NB7-1 wild type and mutant embryos. Data was plotted as described in (A–C). For both Pdm in WT (A) and Cas in the Kr mutant (E), early transient induction of the TTF was detected at stage 9. We attribute this to basal transcription levels which cause protein production until enough repressor is accumulated. Interestingly, for Cas no such transient early induction appears in WT and Hb mutant (E, *Figure 5—figure supplement 1*). These results suggest the transient early induction is an Hb-related phenomenon which could stem from slow Hb accumulation, compared to Pdm and Cas. (G) Position of the in vivo timer in the Pdm Hb-Kr sensitivity space: The measured changes in Pdm induction time following Kr and Hb deletions were used to position the in vivo circuit in the Hb-Kr sensitivity space (black diamond), as described for the simulated data *Figure 3B*. Error bars are

*Figure 5 continued on next page*

*Figure 5 continued*

based on the experimental temporal resolution (see Materials and methods for details). (**H**) Estimating the sensitivity of the in vivo timer to removal of delay or relay interactions: The measured changes in Pdm and Cas induction times following TTF deletions were used to position the in vivo circuit on the correlation plots from *Figure 3C–F* by taking the measured range denoted by 1,2 for ΔHb and 3,4 for ΔKr in *Figure 5G* and measuring the corresponding ranges for regulation removal. These are denoted by 1′,2′ and 3′, 4′ respectively in *Figure 5H* upper plots. Similarly, this was done for Cas regulation in *Figure 5H* lower plots (see Materials and methods for details). (**I**) The TTF circuit is positioned in the region of repressor-delay timers: data from A-H above defined the positioning of the TTF circuit on the Decay-Relay timer space. The location of the in vivo circuit on the X axis, indicating the significance of the decay network, was defined by the range between points 1′ and 2′: point 1′ is the lower bound on decay significance and point 2′ is the upper bound- based on Pdm. Together with a corresponding analysis for Cas, these points determined the location and error margin for the in vivo circuit on the X axis. Similarly, the location on the Y axis, indicating the significance of the relay network, was defined by points 3′,4′ together with the corresponding points for Cas (see text and Materials and methods). Location of the in vivo circuit is indicated by a black diamond, with error margins indicated by dashed error bars. (**J**) Our analysis indicates that the progress of TTF expression is dominated by repressor-decay, and not activator-accumulation. The NB TTF timer regulatory network is shown again, with dominant decay regulations in bold.

DOI: https://doi.org/10.7554/eLife.38631.010

The following figure supplements are available for figure 5:

**Figure supplement 1.** Cas levels in Hb mutant NB7-1.

DOI: https://doi.org/10.7554/eLife.38631.011

**Figure supplement 2.** Predicted model parameter values.

DOI: https://doi.org/10.7554/eLife.38631.012

Therefore, while Hb is an activator of Kr, it affects Kr expression not by determining its induction time but rather by determining shut-off time, allowing Kr decay only when Hb decays below a threshold, again implementing a repressor-decay, rather than an activator-relay timer. Hb is rapidly degraded during early embryogenesis, with an estimated half-life of ~15 min (*Okabe-Oho et al., 2009*). While this half-life was not measured directly in NBs, it is likely to be similarly short based on the 1:1 relationship between *hb* transcriptional activity (detected with an intron probe) and Hb protein levels (detected with an antibody) (*Grosskortenhaus et al., 2005*). Assuming that Hb mRNA is similarly fast degrading, and that Hb is transcribed for only one cell-cycle, we estimate that Pdm starts expressing when Hb levels reduce to about 1–10% of their maximal value.

Our quantification of Pdm levels in wild-type embryos revealed that this phase is longer than previously thought, and led to the identification of a previously unrecognized Kr +Pdm + GMC generated during this phase. Six motor neurons have previously been reported for the NB7-1 lineage (*Landgraf et al., 1997*; *Schmid et al., 1999*), yet there are only five Eve + motor neurons in the lineage, raising the possibility that this 'new' GMC may produce an Eve-negative motor neuron. Our analysis also revealed that Pdm is expressed in a burst during the Hb window. This was noted by *Isshiki et al. (2001)* but neither the functional significance nor the mechanism was discussed. Regarding function, we suggest that this early window of Pdm expression may allow it to be inherited in the first-born GMC, where in at least one lineage (NB4-2) it is required to specify first-born GMC identity, together with Hb (*Bhat et al., 1995*; *Bhat and Schedl, 1994*; *McDonald et al., 2003*; *Yang et al., 1993*; *Yeo et al., 1995*). Regarding mechanism, early Pdm and Cas expression is likely to be due to independent transcriptional activation of both genes, followed by repression of *pdm* transcription by Hb protein (*Kambadur et al., 1998*). The Pdm protein produced from the initial transcriptional burst, prior to Hb-mediated transcriptional repression, may persist into the new-born GMC. Alternatively, there may be a mechanism for blocking Hb repression of *pdm* transcription specifically in early-forming NBs.

We propose that repressor-decay dominating TTF progression was favored in evolution because it better buffers variation ('noise') in molecular parameters. Robust TTF progression is needed to maintain synchrony with the NB division cycles, a prerequisite for generating a reproducible NB lineage. At first sight, repressor degradation and activator accumulation may appear equivalent for measuring time delays. However, closer examination shows that they are in fact very different (*Rappaport et al., 2005*). First, activator accumulation requires continuous transcription while repressor degradation occurs following transcription shutdown. Second, activator accumulation approaches some steady state, which limits the possible readout thresholds. Furthermore, most of the dynamics is spent close to this threshold, so that small changes in threshold levels are translated into large changes in the measured delay time. By contrast, there is no such (theoretical) restriction

on the readout threshold as a repressor decays to zero expression. Together, these properties lead to different buffering capacities, both when considering a single-step timer, and in the context of the full TTF timer model. In all cases, encoding time-delays by repressor decay greatly promotes robustness.

In this study, we did not measure robustness experimentally and did not verify experimentally the predicted difference in the robustness of relay- and decay-based timers. Such a comparison can be done by identifying biological circuits that encode these two timers and are equivalent in all other aspects. Alternatively, the respective timers can be engineered using a synthetic biology approach.

Testing experimentally the proposal that robustness acts as a selection pressure may be more difficult. Defining selective pressures that act during evolution is intrinsically complicated by our limited knowledge of species natural history. The hypothesis that robustness serves as an evolutionary-relevant selection pressure proved useful in defining the evolutionary selected circuit designs in multiple studies from our lab and from others (*Alon et al., 1999*; *Barkai and Leibler, 1997*; *Eldar et al., 2002*; *Eldar et al., 2003*; *Gavish et al., 2016*; *Hart et al., 2014*; *Haskel-Ittah et al., 2012*), and our current study further supports this notion. Still, experimentally, this notion remained untested.

In conclusion, we propose that the need to maintain robust gene expression timing within a noisy biological environment favored evolution of repressor-decay regulatory circuits controlling developmental patterning. This was previously shown for circuits that coordinate spatial patterning through the establishment of morphogen gradients or the control of direct cell-to-cell communication (*Barkai and Shilo, 2009*; *Eldar et al., 2002*; *Eldar et al., 2003*; *Gavish et al., 2016*; *Rahimi et al., 2016*). Our study suggests that robustness also played a major role in the design of developmental timers that function in neuronal differentiation.

## Materials and methods

### Computational methods

#### Mathematical model and numerical screen

Randomized parameter sets (circuits) were generated by randomly drawing values for model parameters out of the ranges indicated in *Supplementary file 4*, using a log uniform distribution. Parameters were then substituted into model equations (*Supplementary file 1*) and solved numerically by a standard MATLAB ODE solver. We selected the ranges for model parameters as follows. First, we chose a 'reference' set of parameters, which gave rise to a consistent circuit and are biologically reasonable based on experimental data. The overall time for the simulation was selected based on the duration of the cascade as measured in (*Grosskortenhaus et al., 2006*; *Isshiki et al., 2001*), the degradation rates were based on measurements of Hb half-life of ~15 min (*Okabe-Oho et al., 2009*) which corresponds to a degradation rate of ~0.05 [1/min] – which is the middle of the range we used for degradation rates. While this half-life was not measured directly in NBs, it is likely to be similarly short based on the 1:1 relationship between hb transcriptional activity (detected with an intron probe) and Hb protein levels (detected with an antibody) (*Grosskortenhaus et al., 2005*). Degradation rates for the other TTFs were selected to be close to this value. Since production rates and thresholds are only significant as a ratio in our model (*Supplementary file 1*), their units can remain arbitrary as long as biologically reasonable ratios are maintained. Since we estimate that Pdm starts expressing when Hb levels reduce to about 1–10% of their maximal value (*Grosskortenhaus et al., 2006*) and a threshold of 1% was used in our previous analytical analysis (*Rappaport et al., 2005*) we restricted ourselves to this range of ratios between maximal levels (ratio between production and degradation) and thresholds. Second, for each model parameter, the range of possible values was selected to be wide going over several orders of magnitude around the value used for the reference set in order to capture the systems behavior in a large part of parameter space. We believe these ranges are wide enough to capture biologically reasonable values. The model we solve numerically always includes all the known interactions between the TTFs (both the repressor decay and activator relay interactions). The values of these randomly drawn parameters determine the relative significance of the interactions in driving the cascade. By repeating this process for millions of such circuits, we examine a wide range of network designs spanning from circuits which are driven mainly by decay to circuits driven mainly by relay and combinations of both.

## Consistency

The solutions were tested for consistency: a consistent solution is one in which the temporal sequence of 'on' (above threshold) TTFs is according to experimental observations for both WT and all mutants (*Figure 1C,D*). The durations of phases were not considered here, only their sequence.

## Robustness and robustness score

Consistent parameter sets were scored for robustness to TTF production rates. When testing set robustness, we solved model equations for all combinatorial combinations of adding or subtracting 20% to all the TTFs production rates. Since the model contains 6 TTF production rate parameters (TTF regulated production for Hb,Kr,Pdm and Cas and basal production for Pdm and Cas- see *Supplementary file 1*) this yielded $2^6 = 64$ noise combinations. For example, one such noise combination is a parameter set where Hb, Kr,Pdm production rates were each increased by 20% and Cas production and Pdm and Cas basal production were each decreased by 20%. For each noise combination, we compared TTF expression phase durations to those of the original set solution. Only if a noise combination yielded phase durations which are all within 10% distance of the respective original durations, the noise combination was considered 'close' to the original. A robustness score was then calculated as the percentage of 'close' noise combinations. In the example shown in *Figure 2D*, the two Pdm temporal dynamics plots for the perturbed systems #1 and #N, show Pdm levels in a consistent parameter set (solid line) and in the noise combination (dashed line). The expression phase of Pdm is 'close' to the original in #1 and not 'close' in #N. The overall robustness score for this parameter set was calculated using this and all other noise combinations and expression phases not shows in these plots. The expression phases considered for this purpose were expression/co-expression phases leading to *different* neuronal fates (*Supplementary file 2*). For example, a phase of Hb only expression followed by co-expression of Hb and Kr was considered a single phase since both lead to the 1 and 2 neuronal fates rendering the timing of Kr induction irrelevant in terms of NB lineage.

## Perturbed parameter sets: deletion of TTF and specific regulations in the model

In order to create perturbed parameter sets, parameter values or terms in model equations were changed accordingly: for TTF deletion the respective production rates were set to 0. For constitutive expression of a TTF, all the terms in model equations regulating this TTFs production were set to 1. For specific regulation removal, the regulation term was set to 1 (see *Supplementary file 1* for model equations).

## Placing parameter sets in Decay-Relay space

For *Figure 1H*, *Figure 2* and *Figure 2—figure supplement 1*, *Figure 2—figure supplement 2* significance scores of decay and relay for each consistent parameter set were first calculated with respect to Pdm and Cas separately (*Figure 2—figure supplement 1*) and then a combined score based on both TTFs was calculated (*Figure 1H*, *Figure 2*, *Figure 2—figure supplement 2*). The TTF-specific scores for Pdm and Cas were based on change in Pdm/ Cas induction times caused by removing the decay or relay regulations governing these inductions. For example, for Pdm the decay removal perturbation is removal of Hb–|Pdm, and the relay removal perturbation is the removal of Kr–>Pdm. We then measured Pdm/Cas induction times in the perturbed sets. The difference from WT in induction time for each perturbed set was calculated as $\Delta t^{ind} = \frac{100*\left|t^{ind}_{Perturbed} - t^{ind}_{Wild\ Type}\right|}{t^{ind}_{Wild\ Type}}$. The combined decay significance score, based on both Pdm and Cas, was calculated by summing $\Delta t^{ind}$ for Pdm (perturbation: removal of Hb–|Pdm) and $\Delta t^{ind}$ for Cas (perturbation: removal of Kr–|Cas). The relay significance score was calculated similarly, only with relay removal perturbations for both Pdm and Cas. The possible range (in absolute value in minutes) of earlier induction or delay is different between Pdm and Cas. For example, Pdm could potentially be up-regulated at t=0 when its repressor Hb is removed. For Cas, upregulation at t=0 is not possible because even when the repressor Kr is absent, another repressor Hb is still active. Due to this difference, when calculating the combined delay and decay scores a normalization was performed: each TTF-specific relay/decay $\Delta t^{ind}$ was taken as percentage out of the maximal $\Delta t^{ind}$ observed for all sets.

## Calculating the location of the in vivo system in Decay-Relay space

For Pdm in *Figure 5G*, the calculation of *in vivo* system location according to its $\Delta t^{ind}$ for TTF deletion perturbations was performed by assuming the experimentally observed $t^{ind}_{Wild\ Type}$ and $t^{ind}_{Perturbed}$ were the middle of the stage in which induction occurred. The assumed error margin was half that stage duration. These error margins were further increased when translating from $\Delta t^{ind}$ for TTF deletions to $\Delta t^{ind}$ for appropriate regulation removal using the model. This translation was performed by placing the measured $\Delta t^{ind}$ for TTF deletion on the appropriate correlation plot in *Figure 5H*. This range is shown in **Figure 5H** upper left plot, by the two points denoted 1,2. Points 1 and 2 represent the upper and lower bound on measured earlier induction of Pdm due to Hb deletion. This range is then translated by the model into the possible range of earlier Pdm induction due to removal of the Hb—| Pdm regulation. This is done using the same correlation plot, by taking the maximal range on the Y axis reached by robust (robustness score>80) sets, within the X axis range of the measured $\Delta t^{ind}$ as defined by points 1,2. This results in the range defined by points 1' and 2' which define the range for $\Delta t^{ind}$ caused to Pdm upregulation by Hb–|Pdm removal. Similarly, points 3,4 indicating the lower and upper bound on Pdm $\Delta t^{ind}$ due to Kr deletion were translated to points 3' and 4' indicating the lower and upper bound on Pdm $\Delta t^{ind}$ due to the removal of Pdm activation by Kr (*Figure 5H* upper right plot). A similar analysis was carried out for Cas, in *Figure 5H* lower plots.

In *Figure 5I*, experimentally measured values of the decay-relay significance scores were indicated on *Figure 2E* by a black diamond shaped marker. These experimental, joint Pdm and Cas, decay and relay significance scores were calculated as previously described for simulated parameter sets: by adding the scores for both Pdm and Cas and then normalizing. Normalization of measured $\Delta t^{ind}$ for Pdm was done by dividing by greatest experimentally observable values: Pdm induction at t=0 which corresponds to $\Delta t^{ind} = 100\%$ in the decay case and the middle of the last stage in the experiment (S12) in the relay case. For Cas, there was no need to normalize $\Delta t^{ind}$ for relay removal since Cas induction in this case occurred during the last stage (S12) in the experiment. Cas decay removal $\Delta t^{ind}$ was normalized by greatest $\Delta t^{ind}$ observed in simulated parameter sets since we cannot expect Cas upregulation at t=0 due to inhibition by Hb.

## Experimental methods

### Data acquisition

The following flies were used: (1) $y^1w^1$ (FBst0001495); (2) $hb^{FB},hb^{P1}/TM3\ ftz\text{-}lacZ$ (*Isshiki et al., 2001*); (3) $Kr^{CD},Kr^1/CyO\ wg\text{-}lacZ$ (*Isshiki et al., 2001*); (4) $Df(2L)ED773/CyO\ wg\text{-}lacZ$ (*Grosskortenhaus et al., 2006*); (5) $ac\text{-}VP16^{AD},gsb\text{-}Gal4^{DBD}$ (*Kohwi and Doe, 2013*); (6) $w^{1118}$; $10xUAS\text{-}IVS\text{-}myr::sfGFP\text{-}THS\text{-}10xUAS(FRT.stop)myr::smGdP\text{-}HA(attP2)$(FBst0062127). Embryos were collected and incubated at 25°C until designated stages, and then fixed and stained with antibodies by following published protocols (*Grosskortenhaus et al., 2005*; *Kohwi and Doe, 2013*; *Tran and Doe, 2008*). The primary antibodies used in the studies were: rabbit anti-Ase (Cheng-Yu Lee, University of Michigan), mouse-anti-beta-galactosidase (Promega), rabbit anti-Cas (*Mellerick et al., 1992*) (Doe lab), rat anti-Dpn (Abcam, Eugene, OR), mouse anti-En 4D9 (Developmental Studies Hybridoma Bank (DHSB), Iowa City, IA), mouse anti-Hb (Abcam, Eugene, OR), guinea pig anti-Kr (Doe lab), rat anti-Pdm2 (Abcam, Eugene, OR), and rabbit anti-Wor (Doe lab). Fluorophore-conjugated secondary antibodies were from Jackson ImmunoResearch. Confocal images were taken by Zeiss LSM710 and protein quantities were measured with open software FIJI (*Schindelin et al., 2012*).

### Data analysis

Data was processed and plotted in MATLAB. Mean volume and standard deviation (STD) for all wild type NB7-1s were calculated. All NBs whose volume was further than 2 STD from mean volume (above or below) weren't included in analysis, assuming these are currently dividing or miss identified cells.

### Determining TTF induction stage in WT and mutants

In order to determine the stage of Pdm and Cas induction from box plots in *Figure 5A–F,a* 'background' level for both TTFs was defined as the mean of their mean levels (red line in box plots) in the second and third stages (S10E and S10), for WT. The first stage (S9) was not considered for this

purpose due to possible early transient induction which occurs for Pdm. The stage of induction was then defined as the first stage for which the 1.96 SEM (95% confidence interval- red rectangle) was above this calculated background.

## Acknowledgements

We thank Benny Shilo, Eyal Schejter, Danny Ben-Zvi and Gat Krieger for comments on the manuscript; Cheng-Yu Lee, Ward Odenwald and DHSB for antibodies. S-LL and CQD were funded by the Howard Hughes Medical Institute, NIH R01-HD27056, and BSF 2017055. NB was supported by a grant from the ERC and BSF 2017055.

## Additional information

### Competing interests

Naama Barkai: Reviewing editor, *eLife*. The other authors declare that no competing interests exist.

### Funding

| Funder | Grant reference number | Author |
| --- | --- | --- |
| Howard Hughes Medical Institute | | Chris Q Doe |
| National Institutes of Health | R01-HD27056 | Chris Q Doe |
| European Research Council | | Naama Barkai |
| United States - Israel Binational Science Foundation | 2017055 | Chris Q Doe<br>Naama Barkai |

The funders had no role in study design, data collection and interpretation, or the decision to submit the work for publication.

### Author contributions

Inna Averbukh, Conceptualization, Data curation, Software, Investigation, Writing—original draft, Writing—review and editing, Modeling, data analysis; Sen-Lin Lai, Investigation, Writing—original draft, Writing—review and editing, animal experiments, data analysis; Chris Q Doe, Naama Barkai, Supervision, Writing—original draft, Writing—review and editing, helped design experiments

### Author ORCIDs

Inna Averbukh  http://orcid.org/0000-0003-1863-205X
Sen-Lin Lai  http://orcid.org/0000-0002-7531-283X
Chris Q Doe  https://orcid.org/0000-0001-5980-8029
Naama Barkai  http://orcid.org/0000-0002-2444-6061

### Decision letter and Author response

Decision letter https://doi.org/10.7554/eLife.38631.020
Author response https://doi.org/10.7554/eLife.38631.021

## Additional files

### Supplementary files

• Supplementary file 1. Mathematical model of the TTF timer – equations and parameters.
DOI: https://doi.org/10.7554/eLife.38631.013

• Supplementary file 2. Table of neuronal fates Neuronal fates induced by TTF co-expression in the NB. Neuronal fates for co-expression of TTFs in the NB at time of division were deduced from lineages described in *Figure 1D*. For every combination, the resulting fate is specified along with the

genotypes from *Figure 1D* from which fate was deduced. Constitutive expression genotypes are denoted by const and deletions by Δ.

DOI: https://doi.org/10.7554/eLife.38631.014

• Supplementary file 3. Table of number of NB7-1s scored per box.

DOI: https://doi.org/10.7554/eLife.38631.015

• Supplementary file 4. Table of Parameter ranges used when searching for consistent sets. Drawing was done from a log-uniform distribution on indicated ranges. When no specific TTF is indicated for the parameter, it is the same for all four TTFs.

DOI: https://doi.org/10.7554/eLife.38631.016

• Transparent reporting form

DOI: https://doi.org/10.7554/eLife.38631.017

## Data availability

All data generated or analysed during this study are included in the manuscript and supporting files.

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
