## [Decision Letter]

Thank you for submitting your article "A repressor-decay timer for robust temporal patterning in embryonic *Drosophila* neuroblast lineages" for consideration by *eLife*. Your article has been reviewed by three peer reviewers, including Wenying Shou as the Reviewing Editor and Reviewer #1, and the evaluation has been overseen by Aviv Regev as the Senior Editor. The following individual involved in the review of your submission has agreed to reveal their identity: Lea Goentoro (Reviewer #3).

The reviewers have discussed the reviews with one another and the Reviewing Editor has drafted this decision to help you prepare a revised submission.

We thought that the work is interesting. However, major points need to be revised and the paper needs to be decompressed. In particular, the reviewers highlight the need to better understand whether repressor decay is in theory (per Figure 2A,B) more robust than activator accumulation (as per the reviews below). We chose to provide you with the full reviews, as all the reviewers agreed those reflect the key issues to be addressed.

Reviewer #1:

In the *Drosophila* embryo, neural progenitors produce a sequence of neurons whose identities depend on the sequential expression of temporal transcription factors (TTF). Although this process is thought to be driven by a relay of activators, Averbukh et al. proposed that a repressor-decay timer is the main player. To evaluate the relative contribution of activator-relay and repressor-decay, they mathematically modeled the TTF timer network, and predicted that with repressor-decay, induction timing of a transcription factor was more robust (changed less) when, for example, activator synthesis rate was reduced. They reasoned that the timer had evolved to be robust, and thus, repressor-decay (which supports higher robustness) should be important. They then did more modeling to design experiments and predict experiment results. Experimental tests supported the repressor-decay mechanism in that the induction timing of Pdm and Cas expression was sensitive to the deletion of the respective repressor and much less so to the deletion of activator. The followings are NOT tested experimentally: repressor-decay timer being more robust than activator-relay timer, and robustness being the selective pressure for evolving repressor-decay timer. However, I do like how the authors use modeling in different ways to gain biological insight and to instruct experiment design. Modeling worked rather well!

The biggest problem I have with this paper is the argument of repressor-decay causing less perturbation in induction timing than activator-accumulation (Figure 2A-B). Seems that this assertion is sensitive to the line slope (the less steep a line e.g. Figure 2A, the bigger the timing perturbation). The line slope will in turn depend on parameters, and the two mechanisms can have different parameters. The argument on this point is also rather hand-wavy in the Discussion (second paragraph). This needs to be clarified.

I find Figure 3B difficult to follow. If my understanding is correct, then the lower right dot essentially says that for this particular set of parameters, deleting HB (but not Kr) causes dramatic phenotype, meaning that HB (decay timer) is important. With this set of parameters, the original "wildtype" network is robust. It took me a couple readings to get this. This needs to be explained better.

"Robustness score", a concept key to this article, had no visual aids. Even in the Materials and methods, the explanation of robustness score was not clear (e.g. "phase duration"). I recommend adding conceptual illustrations like Figure 2D.

Reviewer #2:

I find all the exercise of constraining the model with data quite interesting. I also find the experimental results interesting. That said, I am not entirely convinced by the logic of the reasoning presented. For instance, the experimental results presented in the fourth paragraph of “Timing of Pdm and Cas expression is highly sensitive to deletion of TTF repressors, but less sensitive to deletion of TTF activators” validate the idea that the system is not a relay timer. But do we really need the theoretical study on robustness to get there? In fact, to validate the repressor decay vs. the activator relay model, the only solution is to directly perform those experiments (and maybe other ones to really validate the mechanism). The paper tries very hard to argue that an elaborated theory related to robustness is needed to predict the network topology, but I am rather unconvinced. It could be that the activator relay mechanism is impossible for other reasons that have nothing to do with robustness, so such robustness arguments are in my opinion neither very illuminating nor conclusive.

The attempt to "force" the model to predict experimental results also leads to the strange third paragraph in subsection “Timing of Pdm and Cas expression is highly sensitive to deletion of TTF repressors, but less sensitive to deletion of TTF activators”, where we basically learn that, after experimental verification, all the calibration of the model related to Pdm is incorrect, but that does not matter. It seems to me that in such situation, it would be more reasonable to use this information to redo the theoretical study with the new calibration; one could well learn something new.

On top of that, I found the paper at times difficult to follow. The paper seems to have been initially written for a journal with a very compressed format, but I believe it would be much better if some details and more explanations were given in the main text (I give some suggestions below but they are not exhaustive).

Other comments (in no particular order):

1) The authors postulate a dichotomy between activator-relay and repressor decay. This seems a bit arbitrary to me. One could well imagine more complex networks, a mix of the two via genes that are not known to be implicated, etc. I understand there is a limit to what one can do on the theory side, but I feel some discussions should be added. For instance is it known that the genes studied in the model are necessary and sufficient for the entire process?

2) I found the introduction of the parameter exploration a bit too concise. It would be good to explain how the parameters were chosen and constrained. For instance are there experimental data that are constraining them like degradation rates? More generally, are the parameters found after optimization consistent with what is known or reasonable?

3) Obviously there are also predictions on the possible ranges of parameters when theory is combined with experimental data. I found this is a potentially very interesting aspect of the paper that is not explored sufficiently. For instance can we get more information on parameters from the experimental constraints shown on Figure 5 G and I?

4) I find statements in the first paragraph of subsection “A TTF circuit can be positioned in the relay-decay timer space based on TTF-deletion phenotypes” on the connections between robustness and evolution too speculative and in my opinion confusing.

Reviewer #3:

In this paper, the authors combined modeling of interactions between four Temporal Transcription Factors (TTFs) with experiments to understand the architecture of the TTF timer in neuroblasts. The authors developed a computational framework to distinguish between relay and decay timers, identified all possible circuits that can reproduce normal and perturbed neuroblast phenotypes, and then showed how delay timers are more robust to parameter variations than relay timers. Lastly, they collected high-resolution data for TTF induction time in WT and knockout fly lines, and concluded that temporal TTF activation is primarily governed by a decay-timer circuit.

The authors concluded that robustness is not only a feature of spatial patterning, but also temporal patterning in embryos. The work also provides a wonderful insight into the longstanding question of activation vs. repression in biology.

Major comments:

1) It is unclear why only production rates are being varied for the perturbation analysis, especially since the reasoning for robustness in decay timers (subsection “A repressor-decay timer is more robust than an activator-relay timer”) is based on sensitivity to thresholds Tr. Can the authors provide a rationale? What happens if other parameters are varied as well?

2) Figures 2E, 3B, and Figure 2—figure supplement 2 may appear conflicting. Figure 3B clearly demonstrates that decay timers are more robust than relay timers, with robust circuits concentrated in the bottom right of the perturbation-space. In Figure 2E however, robustness seems to be dependent on the decay interaction, while invariant to the relay-interaction. Finally, in Figure 2—figure supplement 2, robustness shows more complicated dependencies.a) It would be helpful to see Figure 2E and 5I split up into two figures, one for Pdm induction, and one for Cas induction – possibly as supplemental figures. Combining the two as they are currently in Figure 2E, raise questions if there are some patterns that are missed, especially since Figure 3B and Figure 2—figure supplement 2 look so distinct. Could it be that Pdm, but not Cas, induction is the sensitive step in the network where robustness analysis can distinguish relay vs decay?b) Additionally, the Materials and methods indicate that when estimating the significance of the decay network for Cas induction, only the Kr-Cas interaction is removed, and the Hb-Cas interaction is left intact. Can the authors discuss why the dual-repression of Cas is not needed?

3) Figure 5A-F clearly demonstrate that removing the immediate activators has no effect on Pdm and Cas induction timing, and removing the repressor clearly affects timing of Pdm induction. But we have the most trouble with Figure 5E.a) First, Cas induction is pretty modest in both at st11 and 12. Then, based on the network, Cas represses Pdm, and we see this borne out in WT, where at st12, high Cas correlates with low Pdm. However, in Kruppel mutant, Pdm remains high, which seems to signify that there isn't much Cas induction? Can the authors discuss how they see these data? Is there an independent way to confirm that Cas is induced, and induced earlier?b) For Figure 5A-F: It would be helpful to draw a line indicating the "background level" of TTFs, to allow readers to see significance more easily. It would also be helpful to immediately see in the legend the way significant induction is determined. Also, the black arrows are not defined in legend.c) Cas is also repressed by Hb. Can the authors justify why they didn't analyze Cas induction in Hb mutant?

4) It would help to have the Materials and methods be better organized. Perhaps with separate sections, so readers can easily find the relevant information. For instance, we had trouble keeping track of the different ways *Δt^ind^* normalization was performed.

[Editors' note: further revisions were requested prior to acceptance, as described below.]

Thank you for resubmitting your work entitled "A repressor-decay timer for robust temporal patterning in embryonic *Drosophila* neuroblast lineages" for further consideration at *eLife*. Your revised article has been evaluated by Aviv Regev (Senior Editor), and three reviewers, one of whom is a member of our Board of Reviewing Editors.

The manuscript has been greatly improved but there are some remaining issues that need to be addressed by writing changes before acceptance. In particular, the reviewers requested that certain aspects be written more explicitly and clearly, and in a manner geared toward a general audience. Furthermore, they ask that the evolution aspect should be de-emphasized given the lack of direct data, although raising all such matters in the Discussion section should address this concern.

Because some of the reviewers have been grappling in their consultation specifically around writing and presentation, we highlight below the key areas and items that need to be addressed, and the specific writing revisions we request.

1) One reviewer still found the paper rather hard to read, and was not really convinced by the articulation between theory and experiments as is. The experiments done directly show some repressions from upstream genes, so as pointed out in the first review, one does not really need a very elaborated theory to predict this since it is a direct verification. The understand the authors argument that theory helps better refining what experiments to do, but believe that the authors should explain this more clearly. The actual experiments potentially related to theory are in Figure 5: they are connected to the correlations between times on Figure 3 C-D. I found the explanations there too short and concise, and unclear. There is too much of handwaving ("as seen in") making the arguments difficult to understand. On the one hand there are many mutants, on the other hand each mutant gives essentially one point in parameter space that is then placed in the abstract sensitivity space. There are several layers of reasoning here that could be much better explained (Figure 5H is particularly obscure). The reviewer also felt there could be some intuitive or analytical explanations. The authors allude to some analytical work in their rebuttal letter, is there a simple way to interpret the tight correlations of Figure 3 (it seems the correlation simply comes from the existence of one activation)? Also it seems the only argument for the "decay" part is the point 1', but is this really a strong effect?"

To address this, the reviewers together suggest:

The paper should clarify that it presents two parallel arguments for the decay timer:

1) The robustness from modeling analysis.

2) The experiment.

(i.e. rather than a "linear" model-predicts-experiment paper). A paragraph in the Introduction to better clarify the logic of the paper could help.

2) One of the reviewers has an ongoing concern with the robustness argument presented as the core of the paper, as hypothetically, there could be many ways to have a more "robust" network to noise. The fact that a less complicated network is less robust was not fully convincing in implying that robustness is a good biological criterion to assess the evolutionary origin of the network architecture.

We suggest that the results and interpretation of the paper should stand independent of this conjecture to focus on the repressor decay mechanism as a better explanation of the experimental results (which would roughly correspond to what is done in Figure 5 and associated theory), and reduce overinterpretation of the evolutionary origin of the network structure. Overall, given that the paper does not show that robustness is the selective pressure for evolving repressor-decay timer, we prefer this emphasis be reduced, for example, by moving this point to the Discussion.

3) The Discussion should also include the responses (from the authors' rebuttal) on work that was not done.

4) Another concern from the initial reviews was the issue of predicting parameter values from the simulations, and the authors' response that they could not really see anything. If the parameters are truly completely random in the region compatible with data, we would ask to show it explicitly. It seems a bit paradoxical that, following the authors' line of thought, one could predict so carefully the existence of extra negative interactions from the study, but nothing on the actual parameters corresponding to those interactions. At the very least the negative interactions should have parameters significantly different from a "default" state where they would not contribute. This point should be clarified.

5) While we very much appreciate the extra experiment to address our most important concern about the Cas experiments, we are however, still concerned by the fact that Cas delay-relay space (Figure 2—figure supplement 2B) does not support the robustness argument (third paragraph of subsection “A repressor-decay timer is more robust than an activator-relay timer”). We do see the robustness argument with the Pdm space (Figure 2—figure supplement 2A) and when considering the combined Pdm-Cas space in Figure 2E. This concerns us because it can compromise the overall robustness argument, in several ways:a) Perhaps Figure 2E sets up expectations about the robustness of the decay circuits, only to find out in the supplement that it is more true for Pdm, but not as much for Cas.b) It could also raise doubts on the analysis. E.g., what if Cas is less pronounced because the metric "decay significance" vs. "relay significance" does not capture the effects comprehensively enough for Cas. For instance, Cas is inhibited by Hb and Kr, but one could also view Hb's inhibition of Pdm is an inhibition of Cas (which is not removed in the analysis).

In either case, we ask that this be addressed by some re-writing, e.g., for point a, state earlier the difference between Pdm and Cas analysis, so readers are not led to expect more after seeing Figure 2E.

For point b: Add more explanation to the label "significance of decay", perhaps in parentheses, with what is actually being evaluated, e.g., "removal of Pdm--|Cas interaction".

---

## [Author Response]

We thought that the work is interesting. However, major points need to be revised and the paper needs to decompressed. In particular, the reviewers highlight the need to understand better whether repressor decay is in theory (per Figure 2A,B) more robust than activator accumulation (as per the reviews below). We chose to provide you with the full reviews, as all the reviewers agreed those reflect the key issues to be addressed.

We explain this in some detail below: our theoretical analysis indeed shows that a timer that progresses through decay of repressors is more robust than a timer that progresses by the accumulation of activators. This point is important, and we thank the reviewers for pointing out that our explanation was missing. We now expand on it at length in the revised manuscript

Reviewer #1:In the Drosophila embryo, neural progenitors produce a sequence of neurons whose identities depend on the sequential expression of temporal transcription factors (TTF). Although this process is thought to be driven by a relay of activators, Averbukh et al. proposed that a repressor-decay timer is the main player. To evaluate the relative contribution of activator-relay and repressor-decay, they mathematically modeled the TTF timer network, and predicted that with repressor-decay, induction timing of a transcription factor was more robust (changed less) when, for example, activator synthesis rate was reduced. They reasoned that the timer had evolved to be robust, and thus, repressor-decay (which supports higher robustness) should be important. They then did more modeling to design experiments and predict experiment results. Experimental tests supported the repressor-decay mechanism in that the induction timing of Pdm and Cas expression was sensitive to the deletion of the respective repressor and much less so to the deletion of activator. The followings are NOT tested experimentally: repressor-decay timer being more robust than activator-relay timer, and robustness being the selective pressure for evolving repressor-decay timer. However, I do like how the authors use modeling in different ways to gain biological insight and to instruct experiment design. Modeling worked rather well!

We thank the reviewer for the support. We agree with the description above of what our study shows and what it does not.

Comparing experimentally the robustness of the two timers: this could be done in two ways, one would require constructing these timers using a synthetic biology approach, namely, engineering cells to express appropriately wired TTFs complying with the two models. Another approach could be conclusively identifying biological circuits that encode the two timers and are equivalent in all other respects. At this time, the former approach is more realistic, and we agree it is a good experiment, which we will contemplate doing. However, it is beyond the scope of the present manuscript.

Showing experimentally that robustness acts as a selection pressure: defining selective pressures that act during evolution is intrinsically difficult, as it is unclear how to simulate the species natural history. Our approach for that is to examine the consequences of evolutionary-plausible hypothesis, and examine whether these hypothesis lead to useful, and experimentally testable predictions about present-day mechanisms. A long-term hypothesis that guides multiple studies in our lab and others is that robustnesspresents an evolutionary-relevant selection pressure that plays a critical role in defining the selected circuit design. Our finding that the computationally identified robust design correctly predicts the in-vivocircuit mechanism supports this hypothesis and approach.

The biggest problem I have with this paper is the argument of repressor-decay causing less perturbation in induction timing than activator-accumulation (Figure 2A-B). Seems that this assertion is sensitive to the line slope (the less steep a line e.g. Figure 2A, the bigger the timing perturbation). The line slope will in turn depend on parameters, and the two mechanisms can have different parameters. The argument on this point is also rather hand-wavy in the Discussions (second paragraph). This needs to be clarified.

We thank the reviewer for this question, which made us realize that we did not explain well enough how we compared the two mechanisms. As stated by the reviewer, the performance of each model depends on the specific kinetic parameters quantifying the respective reactions, and it is therefore not possible to provide one number defining model’s robustness: within each model,

robustnessdepends on the choice of parameters. This raises the fair question of whether it is even possible to compare the robustness of two models.

Our approach for comparing the robustness of different models (e.g. Figure 2A-B) is to control for the quantitative function of the circuit. In our context, we compare two circuits, implementing the respective timer mechanisms, using parameters that result in the same absolute time-delay. This restricts the value of parameters within each model, and allows for a fair and controlled comparison. Specifically, we base our controlled comparison on the following:

1) For some parameters (e.g. protein degradation time), the identity of the factor, as an activator or a repressor, does not matter. We do not change these parameters between the circuits. That is, degradation rates are the same in compared circuits.

2) Other parameters do change between the models. For example, the threshold for TF repression could be different from the threshold for TF activation in order to maintain the same time delay for both models. In comparing the models, we choose all these parameters in a way that ensures that the two models define the same time-delay.

Our general, theoretical comparison of the one-step timer is done for circuits that comply with conditions 1-2. The results are shown in Figure 2. Of note, in this simple case, we can solve the system analytically and by this show that for any parameter choice complying with conditions 1-2, the repression-decay is more robust than activator relay.

For a longer cascade, we performed a similar comparison computationally. Figure 2E, for example, compares robustness of all circuits that comply with the experimentally-defined dynamics. We do note that in this case, we do not demand a precise pattern of time delays, but do allow some range of possible dynamics, to account for the low precision of the experimental data. Therefore, not all circuits in this comparison show the precise same dynamics, but they all fall within a similar range that is compatible with the experimental data. We present our data in this way for simplicity. Increasing the precision, namely comparing only circuits that show the precise same time delays, does not change the results: decay-based timers are more robust than relay-based ones. Therefore, our statement that the repressor-decay timer is more robust than the activator-relay one is indeed general for all ‘fair’ comparisons that comply with conditions 1-2.

We now explain this in the manuscript, and add further analysis substantiating our claims (Figure 2—figure supplement 1).

1) When explaining the results of the general model:

“Comparing the robustness of two models is convoluted by the choice of parameters: indeed, similar to the absolute time delays, robustness (defined by sensitivity to parameter variations), is a quantitative measure that depends on the kinetic parameters. Still, the two models can be compared by controlling for the absolute time delays. Thus, we compare activator and repressor that (1) have the same decay rate and (2) require the same time transitioning between two given thresholds. As we showed, the repressor-decay timer necessarily shows higher robustness under these conditions (Rappaport et al., 2005).”

2) When explaining the results of our simulations:

“To rigorously distinguish whether robustness correlates with a specific timer type, we considered again the positioning of all circuits in the decay-relay timer space (c.f. Figure 1H). Unlike the simple case of the one step timer, here we did allow some range of possible dynamics, to account for the low precision of the experimental data. This does not affect our ability to compare the consistent circuits since these differences in dynamics are not correlated with location in relay-decay space (Figure 2—figure supplement 1).”

I find Figure 3B difficult to follow. If my understanding is correct, then the lower right dot essentially says that for this particular set of parameters, deleting HB (but not Kr) causes dramatic phenotype, meaning that HB (decay timer) is important. With this set of parameters, the original "wildtype" network is robust. It took me a couple readings to get this. This needs to be explained better.

Yes – this is what the figure shows. We now explain it better in the text:

“With this in mind, we examined computationally whether the consequences of TTFs deletions, if analyzed at higher resolution, could distinguish between the repressor-decay and activator-relay timers. […]Once positioned, we color each circuit by its robustness score for the WT circuit (same color as in Figure 2E) which allows us to observe robustness of WT circuits as function of their Pdm induction sensitivity to Hb and Kr deletions (Figure 3B).”

And in the caption of Figure 3:

“(A-B) TTF deletion phenotypes can distinguish robust circuits: all consistent circuits, as described in Figure 1-2 above, were considered. […] This analysis shows that robust circuits are only found in a small region in the Kr-Hb sensitivity space, in which Pdm induction time is much more sensitive to Hb than to Kr deletion.”

"Robustness score", a concept key to this article, had no visual aids. Even in the Materials and methods, the explanation of robustness score was not clear (e.g. "phase duration"). I recommend adding conceptual illustrations like Figure 2D.

We thank the reviewer for this suggestion. We replaced Figure 2D with a panel which better explains the concept and added a clearer description of what we do to the Materials and methods section:

“Robustness and robustness score

Consistent parameter sets were scored for robustness to TTF production rates. […] For example, a phase of Hb only expression followed by co-expression of Hb and Kr was considered a single phase since both lead to the 1 and 2 neuronal fates rendering the timing of Kr induction irrelevant in terms of NB lineage.”

Reviewer #2:I find all the exercise of constraining the model with data quite interesting. I also find the experimental results interesting. That said, I am not entirely convinced by the logic of the reasoning presented. For instance, the experimental results presented in the fourth paragraph of “Timing of Pdm and Cas expression is highly sensitive to deletion of TTF repressors, but less sensitive to deletion of TTF activators” validate the idea that the system is not a relay timer. But do we really need the theoretical study on robustness to get there? In fact, to validate the repressor decay vs. the activator relay model, the only solution is to directly perform those experiments (and maybe other ones to really validate the mechanism).

We appreciate this comment and would like to answer it on two levels. First, as we are sure the reviewer acknowledges, the space of possible experiments is large, and defining the experiment that will provide maximal information about the underlying mechanism is difficult. Modeling can therefore direct experimental efforts to the most informative measurements, and define the resolution needed, as was the case in the present study. It is indeed the fact that these experiments were motivated by the theoretical analysis.

Second, theory is instrumental for comparing qualitative properties of different mechanisms. In the present case, one could naively assume that activator-relay and repressor-decay timers are equivalent. Indeed, mechanistically, both can

define proper timing. Our study showed, however, that the two mechanisms are very different in terms of robustness. At least for us, this qualitative difference was the main motivation to experimentally test which of these mechanisms guides the in-vivo timer.

The paper tries very hard to argue that an elaborated theory related to robustness is needed to predict the network topology, but I am rather unconvinced. It could be that the activator relay mechanism is impossible for other reasons that have nothing to do with robustness, so such robustness arguments are in my opinion neither very illuminating nor conclusive.

We appreciate this comment, and of course cannot refute the possibility that an unknown constraint made it impossible to implement the relay-based timer in-vivo. Our computational analysis suggests that activator-relay can define a timer, but that it is limited in robustness. So all we can say is that the mechanism used in-vivoconforms to the robust design. This corroborates our conjecture that the need for robustness restricts the design of biological circuits, but of course does not prove it.

The attempt to "force" the model to predict experimental results also leads to the strange third paragraph in subsection “Timing of Pdm and Cas expression is highly sensitive to deletion of TTF repressors, but less sensitive to deletion of TTF activators”, where we basically learn that, after experimental verification, all the calibration of the model related to Pdm is incorrect, but that does not matter. It seems to me that in such situation, it would be more reasonable to use this information to redo the theoretical study with the new calibration; one could well learn something new.

We thank the reviewer for this comment – perhaps we did not describe this result clearly enough. It is not that the initial calibration was wrong, but it was of an insufficient temporal resolution. This comment therefore relates again to the issue of defining the most informative experiment. As mentioned above, the space of possible experiments is very large. For this reason, experiments that existed when we began our analysis were mostly of low temporal resolution and the data that we had in hand provided only partial restrictions of the possible parameters. The initial calibration therefore was not wrong, but simply less restrictive.

Specifically for Pdm, the experimental data we had in calibrating the model did not allow us to accurately define its expression period; therefore all solutions that showed proper ordering of the respective TTFs were accepted as consistent. Having this information following our focused time-resolved analysis allowed us to further restrict the analysis, but all the results that were consistent with the new, more restrictive data were also deemed as consistent in the original analysis since we quite deliberately allowed a range of possible dynamics to account for the low temporal resolution of the calibration data.

To better explain that, we added Figure 4—figure supplement 1, where we specifically show the number of Pdm neurons that are defined by the set of circuits that were found to be consistent in the original analysis. As can easily be appreciated form this figure, it includes both the solutions that allow 2-3 Pdm neurons, and these that allow smaller, or larger number.

We also revised the text to better explain that:

“The Pdm expression window is therefore significantly longer than the duration inferred from the previous data used to calibrate our model. This did not affect our analysis, however, since in our original analysis we did allow for solutions of similarly long Pdm windows, accounting for the uncertainty of the experimental data (Figure 4—figure supplement 1).”

On top of that, I found the paper at times difficult to follow. The paper seems to have been initially written for a journal with a very compressed format, but I believe it would be much better if some details and more explanations were given in the main text (I give some suggestions below but they are not exhaustive).

Actually, the paper was not submitted elsewhere – *eLife* is our first choice and the paper was written specifically for this journal. In fact, one of us (CQD) is sending all of the labs “high impact” papers to *eLife* instead of CSN. We do thank the reviewer for the suggestions to clarify different aspects of the analysis, all of which we tried to implement.

Other comments (in no particular order):1) The authors postulate a dichotomy between activator-relay and repressor decay. This seems a bit arbitrary to me. One could well imagine more complex networks, a mix of the two via genes that are not known to be implicated, etc. I understand there is a limit to what one can do on the theory side, but I feel some discussions should be added. For instance is it known that the genes studied in the model are necessary and sufficient for the entire process?

All interactions described in our model have experimental support, as described in our series of studies (Isshiki et al., 2001; Cleary and Doe, 2006; Grosskortenhaus et al., 2005; Grosskortenhaus et al., 2006; Cleary and Doe, 2006; Tran and Doe, 2008). Although we cannot rule out additional regulators, it is believed that the four TTFs are the central drivers of the timer, and additional factors, if exist, would play a more peripheral role.

Please also note that we do not postulate a dichotomy between the activator-relay and repressor-decay models, but in fact allow a continuum of circuits that smoothly shift the mechanism between these two extremes. This is what we capture in the decay-relay phase space, displayed in figures 1H, 2E, and 5I. We now describe this more clearly in the text:

“Further, varying parameters qualitatively changes the regulatory network by varying the relative influence of the different interactions on TTF temporal dynamics. This allows us to capture a wide range of networks, ranging from decay dominant ones, through any mixed models to relay dominant networks all the while taking into account all experimentally observed TTFs and cross-regulations (Supplementary File 1).”

And further elaborate in a new Supplementary File 1 section:

“In this model, the relative significance of specific interactions in determining the temporal dynamics, depends on the choice of parameters. […] For example: when β_pdm_^basal^
*>> 𝛽_𝑝𝑑m_*, induction by Kr will be negligible and downregulation of Hb would be dominant. For β_pdm_^basal^
*<< 𝛽_𝑝𝑑m_*, the opposite is true: Kr-dependent transcription of Pdm is much more substantial.”

2) I found the introduction of the parameter exploration a bit too concise. It would be good to explain how the parameters were chosen and constrained. For instance are there experimental data that are constraining them like degradation rates? More generally, are the parameters found after optimization consistent with what is known or reasonable?

We selected the range of analyzed parameters as follows. First, we chose a ‘reference’ set of parameters, which are biologically reasonable based on experimental data. Second, our screen used parameters that are present within three orders of magnitudes of these reference parameters. This is now explained in the Materials and methods section:

“Mathematical model and Numerical screen

Randomized parameter sets (circuits) were generated by randomly drawing values for model parameters out of the ranges indicated in Supplementary File 4, using a log uniform distribution. […] We believe these ranges are wide enough to capture biologically reasonable values.”

3) Obviously there are also predictions on the possible ranges of parameters when theory is combined with experimental data. I found this is a potentially very interesting aspect of the paper that is not explored sufficiently. For instance can we get more information on parameters from the experimental constraints shown on Figure 5 G and I?

Unfortunately, the range of possible parameter combinations that are consistent with the experiments is still too large to predict reliably the value of any specific quantitative parameters.

4) I find statements in the first paragraph of subsection “A TTF circuit can be positioned in the relay-decay timer space based on TTF-deletion phenotypes” on the connections between robustness and evolution too speculative and in my opinion confusing.We respectfully disagree. Our working hypothesis, used here and in previous studies is that the need for robustness restricts the design of biological circuits. In the lines the reviewer refers to we clearly state this is a hypothesis. In this study, this hypothesis was indeed useful and helped us correctly predict circuit design.Reviewer #3:1) It is unclear why only production rates are being varied for the perturbation analysis, especially since the reasoning for robustness in decay timers (subsection “A repressor-decay timer is more robust than an activator-relay timer”) is based on sensitivity to thresholds Tr. Can the authors provide a rationale? What happens if other parameters are varied as well?

In our formulation, perturbing the thresholds (K_d_) is equivalent to perturbing TTF production rate. This is because the relevant quantity defining state-transitions is K_d_/[TTF], [TTF] being the TTF concentration. Therefore, randomly varying production rates captures also the consequences of randomly varying the thresholds.

As we write, we do not include the degradation rates in the robustness analysis. This is because degradation rates define the time-scale of timer dynamics, and therefore equally affect both mechanisms. For example, in the case of a one-step cascade, we showed analytically that the delay time is linearly proportion to the degradation rate in both models. Similarly, our numerical analysis confirmed this dependency also in the case of a multi-step cascade.

We explain these points in the manuscript:

“In our model, introducing fluctuations in TTF production rate captures also fluctuations in thresholds values, as circuit function depends on the ratio of TTF levels to their activation thresholds. Finally, degradation rates were kept constant, as they define the time scale of timer dynamics and equally affect all circuits ((Rappaport et al., 2005) and data not shown).”

2) Figures 2E, 3B, and Figure 2—figure supplement 2 may appear conflicting. Figure 3B clearly demonstrates that decay timers are more robust than relay timers, with robust circuits concentrated in the bottom right of the perturbation-space. In Figure 2E however, robustness seems to be dependent on the decay interaction, while invariant to the relay-interaction. Finally, in Figure 2—figure supplement 2, robustness shows more complicated dependencies.

We thank the reviewer for this comment. There are two differences between these figures: first in the type of perturbation used to probe the circuit, and second in how the consequences of these perturbations are measured.

Perturbation type: In Figure 2E, Figure 2—figure supplement 2, we simulate the removal of a specific *interaction* (for example, the ability of Hb to repress Pdm). By contrast, Figure 3B and Figure 3—figure supplement 1 simulate deletion of the *respective TTF* (for example, deleting Hb). As is shown in Figure 3C-F, these perturbations lead to similar, but not identical, effects on the Pdm and Cas induction times.

Perturbation consequences: In Figure 2E, we considered the dynamics of both Pdm and Cas, the two steps in the cascade that are completely circuit-autonomous (not sensitive to external signals). The positioning of each circuit in the decay-relay space is therefore defined by the sensitivities of Pdm and Cas induction times to the deletion of the respective interactions. By contrast, in Figure 3B, Figure 2—figure supplement 2 and Figure 3—figure supplement 1, we consider the two TTFs separately. Thus, in Figure 3B we only measure the timing of Pdm induction, whereas in Figure 3—figure supplement 1 we measure the timing of Cas induction.

Therefore, the three figures present the same set of (consistent) circuits, having the same robustness scores (color-coded, based on the WT circuit), yet these circuits are positioned differently, depending on how the decay-relay space is defined: which perturbation are introduced (removing interaction or deleting a factor), and whether we consider both Pdm and Cas (Figure 2E), only Pdm (Figure 3B, Figure 2—figure supplement 2A) or only Cas (Figure 2—figure supplement 2B, Figure 3—figure supplement 1).

a) It would be helpful to see Figure 2E and 5I split up into two figures, one for Pdm induction, and one for Cas induction – possibly as supplemental figures. Combining the two as they are currently in Figure 2E, raise questions if there are some patterns that are missed, especially since Figure 3B and Figure 2—figure supplement 2 look so distinct. Could it be that Pdm, but not Cas, induction is the sensitive step in the network where robustness analysis can distinguish relay vs. decay?

We thank the reviewer for this suggestion. We added a new supplementary figure (Figure 2—figure supplement 2) showing the two separately. As can be appreciated, the increased robustness of the decay timer holds both steps of the cascade also when analyzed separately.

b) Additionally, the Materials and methods indicate that when estimating the significance of the decay network for Cas induction, only the Kr-Cas interaction is removed, and the Hb-Cas interaction is left intact. Can the authors discuss why the dual-repression of Cas is not needed?

We thank the reviewer for this comment, which pointed us to an inaccurate assumption made in our analysis. When performing our simulation, we assumed that, at the time relevant for Cas induction, Kr is present at much higher concentrations compared to Hb, making it the dominant factor repressing Cas. We therefore only considered sensitivity to Kr repression. Following the reviewer comment, we re-examined this assumption. This showed us that the induction time of Cas is indeed sensitive to the Hb-dependent repression in some of the consistent circuits, as is shown in Author response image 1.

Analysis of Cas induction time sensitivity to the removal of Cas repression by Hb. All consistent circuits, as described in Figures 1-2, were considered. Each consistent circuit was scored by measuring the change in Cas induction times following removal of the decay Hb-|Cas (X-axis) or relay Pdm->Cas (Y-axis) regulations. These values were used to uniquely position each circuit in the Hb-Pdm sensitivity space (Materials and methods). Parameter sets are Color-coded based on their robustness score for the WT circuit, as in Figure 2E </Author response image 1 title/legend>

However, this interaction did not appear to provide additional information about the robustness of consistent circuits since robust circuits are found all along the X-axis. This indicates sensitivity to the removal of Cas repression by Hb does not correlate with robustness or location in decay-relay space. We therefore did not include it in the analysis.

3) Figure 5A-F clearly demonstrate that removing the immediate activators has no effect on Pdm and Cas induction timing, and removing the repressor clearly affects timing of Pdm induction. But we have the most trouble with Figure 5E.a) First, Cas induction is pretty modest in both at st11 and 12. Then, based on the network, Cas represses Pdm, and we see this borne out in WT, where at st12, high Cas correlates with low Pdm. However, in Kruppel mutant, Pdm remains high, which seems to signify that there isn't much Cas induction? Can the authors discuss how they see these data? Is there an independent way to confirm that Cas is induced, and induced earlier?

We agree with the reviewer that the induction of Cas in Kr mutant is lower than that of wild-type. Thus, although significant induction is observed earlier than wild-type (at stage 11 Cas is significantly higher in Kr mutants comparted to wild-type), it remains moderate, so that at stage 12, wild-type embryos express significantly higher levels of Cas compared to Kr-mutant embryos.

Still, these moderate levels of Cas, observed in the Kr-mutants in both stages 11-12, are biologically significant, as they are sufficient to induce the Cas-positive U5 neuron generated at this stage (see data in Figure 1D based on Isshiki 2001)

Further, the moderate levels of Cas likely explain why Pdm is not repressed. The simplest explanation of our Kr-mutant data is that Cas is induced earlier than in wild-type, but to moderate levels that are sufficient for generating the Cas-positive neurons, but are too low for repressing Pdm.

In principle, Cas can be independently verified by directly monitoring the activity of *cas* locus with MS2 coat protein fused GFP (Garcia et al., 2013) or directly observe the synthesis of Cas protein with LlamaTag (Bohmer et al., 2018). Due to the limited time-frame of the revision, we could not perform these experiments, but plan to pursue both of these experiments in the future. Please note that our main conclusion is supported by the clear observation that removal of the relay reaction (Pdm) has no effect on Cas induction time.

We refer to this point in the manuscript.

**“**Cas induction in *Kr* mutants was moderate compared to its induction in WT embryos. This moderate induction is sufficient to induce the Cas-positive U5 neuron, produced in Kr mutants (Isshiki et al., 2001), and may result from its earlier induction, at a time when Hb, its second repressor, is still highly expressed. Consistent with the contribution of Hb to Cas repression, Cas induction was advanced in Hb mutants to stage 11, similarly to Kr mutants (Figure 5—figure supplement 1).”

b) For Figure 5A-F: It would be helpful to draw a line indicating the "background level" of TTFs, to allow readers to see significance more easily. It would also be helpful to immediately see in the legend the way significant induction is determined. Also, the black arrows are not defined in legend.

Added

c) Cas is also repressed by Hb. Can the authors justify why they didn't analyze Cas induction in Hb mutant?

We now performed this experiment (Figure 5—figure supplement 1) and discuss this in main text:

“Consistent with the contribution of Hb to Cas repression, Cas induction was advanced in Hb mutants to stage 11, similarly to *Kr* mutants (Figure 5—figure supplement 1).”

4) It would help to have the Materials and methods be better organized. Perhaps with separate sections, so readers can easily find the relevant information. For instance, we had trouble keeping track of the different ways Δt^ind^ normalization was performed.

We revised the Materials and methods section as suggested.

[Editors' note: further revisions were requested prior to acceptance, as described below.]

[…] To address this, the reviewers together suggest:The paper should clarify that it presents two parallel arguments for the decay timer:1) The robustness from modeling analysis2) The experiment.(i.e. rather than a "linear" model-predicts-experiment paper). A paragraph in the Introduction to better clarify the logic of the paper could help.

We added a paragraph to the Discussion and changed the logic of presentation, as suggested. Specifically, we now present our story as follows:

1) Modeling shows that the molecular circuit defined by interactions between the TTF composing the NB timer, can support two, qualitatively different ‘core’ mechanisms: one that progress the TTF cascade by a relay of activator, and one that progresses the TTF cascade by a decay of repressors.

2) Within the actual circuit, both ‘core’ mechanisms can contribute to timer progression. The relative contribution of each mechanism depends on quantitative values of the kinetic parameters.

3) Theory shows that a timer, which progresses by decay of repressors, buffers parameters variations better than a timer that progresses by a relay of activators. (That is, this timer is more robust).

4) This difference between the robustness of the two timers made it interesting to examine which of these two ‘core’ mechanisms dominates timer progression *in-vivo.*

5) Theory suggests experiments that can best distinguish the relative contribution of these two core mechanisms to the progression of the *in-vivo* timer.

6) Experiments show that the repressor-decay mechanism dominates the progression of the *in-vivo* timer. Therefore, the more robust mechanism dominates the *in-vivo* function.

7) In the Discussion, we raise the possibility that evolution favors the selection of robust mechanisms.

In addition to these changes in presenting our logic, we also tried to simplify the explanations of Figure 5, both in the text and in the figure caption.

2) One of the reviewers has an ongoing concern with the robustness argument presented as the core of the paper, as hypothetically, there could be many ways to have a more "robust" network to noise. The fact that a less complicated network is less robust was not fully convincing in implying that robustness is a good biological criterion to assess the evolutionary origin of the network architecture.We suggest that the results and interpretation of the paper should stand independent of this conjecture to focus on the repressor decay mechanism as a better explanation of the experimental results (which would roughly correspond to what is done in Figure 5 and associated theory), and reduce overinterpretation of the evolutionary origin of the network structure. Overall, given that the paper does not show that robustness is the selective pressure for evolving repressor-decay timer, we prefer this emphasis be reduced, for example, by moving this point to the Discussion.

We account for that by changing our presentation, as detailed in our reply to comment 1 above. We move our discussion of the possible evolutionary connection between robustness and the mechanism of choice to the discussion. Robustness is now presented as a property that differs between qualitatively different ‘core’ models; a difference that motivated experiments to distinguish which of these ‘core’ models dominates timer progression *in-vivo.*

We note that the robust model (*repressor decay)* is not more complicated than the non-robust one (*activator relay).* In fact, they can be encoded by the same number of components and interactions.

3) The Discussion should also include the responses (from the authors' rebuttal) on work that was not done.

Added

4) Another concern from the initial reviews was the issue of predicting parameter values from the simulations, and the authors' response that they could not really see anything. If the parameters are truly completely random in the region compatible with data, we would ask to show it explicitly. It seems a bit paradoxical that, following the authors' line of thought, one could predict so carefully the existence of extra negative interactions from the study, but nothing on the actual parameters corresponding to those interactions. At the very least the negative interactions should have parameters significantly different from a "default" state where they would not contribute. This point should be clarified.

We agree. The only point we tried to make is that other predictions do not correspond to qualitatively different mechanisms, and therefore we found them less informative. But the reviewer is of course correct that our analysis distinguishes the range of possible values that complies with the theory.

We now added a supplementary figure 5—figure supplement 2 in which we present the range of parameters that are compatible with all experimental observations. In this figure, we show the range of parameters tested in our simulation, the subset which were proven consistent with existing experiments, available prior to our study, and then the restricted subset of parameters that remained consistent also following the experiments we performed.

Please also note that our analysis does not predict the existence of extra negative interactions, only the dominance of known negative interactions over the known positive ones.

5) While we very much appreciate the extra experiment to address our most important concern about the Cas experiments, we are however, still concerned by the fact that Cas delay-relay space (Figure 2—figure supplement 2B) does not support the robustness argument (third paragraph of subsection “A repressor-decay timer is more robust than an activator-relay timer”). We do see the robustness argument with the Pdm space (Figure 2—figure supplement 2A) and when considering the combined Pdm-Cas space in Figure 2E. This concerns us because it can compromise the overall robustness argument, in several ways:a) Perhaps Figure 2E sets up expectations about the robustness of the decay circuits, only to find out in the supplement that it is more true for Pdm, but not as much for Cas.b) It could also raise doubts on the analysis. E.g., what if Cas is less pronounced because the metric "decay significance" vs. "relay significance" does not capture the effects comprehensively enough for Cas.

The argument of robustness is in fact valid also for Cas. We agree that this was not properly presented in our previous sup Figure 2A, and we apologize for that. The reason for this suboptimal presentation is that, overall, Cas induction time is less sensitive to noise compared to Pdm induction time (it is controlled by two repressors: Hb and Kr). However, in previous revision, when we added sup Figure 2A, we chose to show it with the same level of noise (20%) used for generating Figure 2E of Pdm robustness.

We now show the same plots, but for increasing noise levels (20%, 30% and 40%). This is shown for both Pdm and Cas (Figure 2—figure supplement 1). As can be appreciated, when examined at higher noise levels, the robustness benefit of the decay timers becomes evident also for Cas.

For instance, Cas is inhibited by Hb and Kr, but one could also view Hb's inhibition of Pdm is an inhibition of Cas (which is not removed in the analysis).

This is not in fact the case in our simulations. The inhibition of Pdm by Hb does not affect Cas directly since by the time Cas is induced, Pdm is already upregulated. Therefore the Hb inhibition of Pdm is removed much before Cas is being induced.

Either case, we ask that this be addressed by some re-writing, e.g., for point a, state earlier the difference between Pdm and Cas analysis, so readers are not led to expect more after seeing Figure 2E.

We now refer to the different noise sensitivity of Cas and Pdm in main text. With respect to robustness, we believe that the new sup figure we show verifies that robustness favors a decay timer also in the case of Cas.

For point b: Add more explanation to the label "significance of decay", perhaps in parentheses, with what is actually being evaluated, e.g., "removal of Pdm--|Cas interaction".

Done, see new Figure 2—figure supplement 1 axis labels.